# DialectGen: Benchmarking and Improving Dialect Robustness in Multimodal Generation

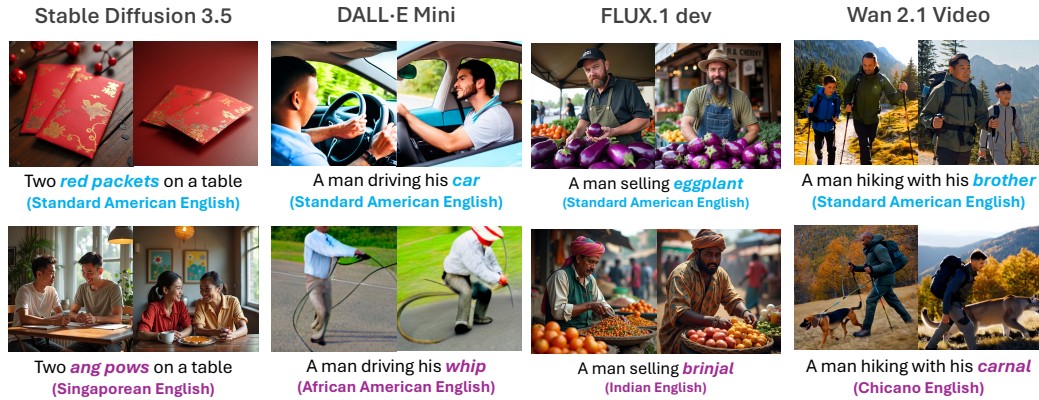

Figure 1: **Multimodal Generative Model Outputs** on semantically identical prompts that differ only in one synonymous lexical feature in Standard American English (top) / a lower-resource English dialect (bottom).

## Abstract

Contact languages like English exhibit rich regional variations in the form of dialects, which are often used by dialect speakers interacting with generative models. However, can multimodal generative models effectively produce content given dialectal textual input? In this work, we study this question by constructing a new large-scale benchmark spanning six common English dialects. We work with dialect speakers to collect and verify over 4,200 unique prompts and evaluate on 17 image and video generative models. Our automatic and human evaluation results show that current state-of-the-art multimodal generative models exhibit 32.26% to 48.17% performance degradation when a single dialect word is used in the prompt. Common mitigation methods such as fine-tuning and prompt rewriting can only improve dialect performance by small margins (< 7%), while potentially incurring significant performance degradation in Standard American English (SAE). To this end, we design a general encoder-based mitigation strategy for multimodal generative models. Our method teaches the model to recognize new dialect features while preserving SAE performance. Experiments on models such as Stable Diffusion 1.5 show that our method is able to simultaneously raise performance on five dialects to be on par with SAE (+34.4%), while incurring near-zero cost to SAE performance.

## 1 Introduction

Linguists have defined over 160 dialects (Aeni et al., 2021) within the English language, with three out of four English speakers having a dialect background other than Standard American or British English (Crystal, 2003). Despite this rich diversity, current pre-training paradigms employ content filters that can exclude data involving lower-resource English dialects other than Standard American and British English (Gururangan et al., 2022), reducing the effectiveness of pretrained models on inputs from other dialects (Lee et al., 2023). Prior works have shown significant allocational harms toward dialect speakers caused by such dialect performance discrepancies in machine learning appli-

cations (Hovy & Spruit, 2016; Bender et al., 2021), making the observation of similar performance trends in multimodal generative models an alarming sign.

As shown in Figure 1, while current multimodal generative models can accurately generate high quality image and video content given Standard American English (SAE) prompts (left); they fail in various manners when provided with semantically equivalent prompts containing a single synonymous dialect word (right). Stable Diffusion 3.5 Large (Esser et al., 2024) fails to generate "ang pow", which is commonly used in Singaporean English to mean "red packet", and FLUX.1 [dev] (Black Forest Labs, 2024) fails to generate "brinjal", which is synonymous with "eggplant" in Indian English. Furthermore, when the dialect lexeme is polysemous, *i.e.*, has an alternative meaning in SAE, models tend to always generate content that align with the SAE meaning, even when the context makes such interpretation highly improbable. For example, DALL-E Mini (Dayma et al., 2021) generations of "A man driving his whip" fail to capture the correct meaning of "whip" as "car" in African American English, given clear context indications. Similar failure modes are observed in text-to-video generative models: Wan 2.1 (Wang et al., 2025) fails to correctly render "carnal", which refers to "brother" in Chicano English.

In this work, we construct **DialectGen**, a new large-scale benchmark evaluating dialect robustness in image and video generation. Our benchmark dataset spans six common English dialects, including Standard American English (SAE), British English (BrE), Chicano English (ChE), Indian English (InE), and Singaporean English (SgE). For each dialect other than SAE, we create SAE Prompt / Dialect Prompt pairs that are semantically identical besides switching a single SAE lexeme for a synonymous dialect lexeme. We work with dialect speaker annotators to create a rigorous feature selection and prompt filtering pipeline that ensures the final dialect prompts are (1) exactly synonymous with the SAE prompt; (2) valid in the dialect context; and (3) non-ambiguous (for polysemous lexemes). These strictly enforced quality guarantees facilitate the development of simple yet effective automatic and human evaluation metrics for evaluating generative model performance. We experiment with 17 widely used image and video generative models on **DialectGen**, demonstrating up to 38.63% and 48.17% performance drops for SOTA open-weights image and video generative models, respectively.

To alleviate such significant dialect performance drops observed in current multimodal generative models, we design a general encoder-based learning strategy that enhances dialect robustness for diffusion-based multimodal generative models. Our method teaches the model's text encoder to recognize dialect lexemes while retaining its knowledge of SAE polysemous lexemes. We also include an encoder-based KL regularization loss based on image-SAE caption datasets to regulate output distribution shifts. Experiments on five dialects show that our method is able to simultaneously improve Stable Diffusion 1.5 (Rombach et al., 2022) and SDXL (Podell et al., 2023) performance on five dialects to be on par with SAE performance. At the same time, we observe near zero (< 1%) SAE performance drop on the general MSCOCO (Lin et al., 2014) validation set for both models.

Our key contributions include:

- **DialectGen**, a new large-scale multi-dialectal benchmark for evaluating dialect robustness in text-to-image and text-to-video generation.
- Comprehensive evaluation and analysis of 17 multimodal generative models and five baseline mitigation methods on **DialectGen**.
- A high-performing method for improving dialect robustness in multimodal generation while maintaining strong SAE performance.

## 2 RELATED WORKS

Linguists define dialects as regional variations of a language distinguished by unique features in lexicon, phonology, and grammar from each other, together constituting a single language (Hudson, 1996; Chambers & Trudgill, 1998; Fromkin et al., 1998; Nerbonne, 2009; Wardhaugh & Fuller, 2021). English, like any other language, is subject to such variations. However, most dataset resources and pre-training paradigms focus only on Standard American and British English (Gururangan et al., 2022), leading to dialect robustness issues and performance gaps in downstream machine learning applications. Previous works have analyzed and explored such dialectal performance gaps in NLP tasks like QA (Ziems et al., 2023), NLI (Ziems et al., 2022), dependency parsing, and POS

Table 1: Example paired textual data entries from the **DialectGen** dataset, including **Lexeme**, **Concise Prompt**, and **Detailed Prompt**. Dialect name abbreviations: SAE (Standard American English), AAE (African American English), BrE (British English), SgE (Singaporean English).

| Dialect | Lexeme | Concise Prompt | Detailed Prompt |
|---------|--------|----------------|-----------------|
| SAE | sneakers | brand new sneakers | a little girl wearing a pair of stylish white sneakers |
| AAE | kicks | brand new kicks | a little girl wearing a pair of stylish white kicks |
| SAE | bathroom | a spacious bathroom | a clean and tidy bathroom with shiny blue wall tiles |
| BrE | loo | a spacious loo | a clean and tidy loo with shiny blue wall tiles |
| SAE | squid | a squid on a counter | a large squid in an aquarium with colorful coral |
| SgE | sotong | a sotong on a counter | a large sotong in an aquarium with colorful coral |

tagging (Blodgett et al., 2018; Jørgensen et al., 2015). Recent works have also noticed the impact of dialect variations on text-to-image generation (Lee et al., 2023; Wan et al., 2024). Along this line of research, we create the first large-scale benchmark of dialect robustness in multimodal generation, evaluating both text-to-image and text-to-video generative models on inputs across six different dialects.

Moreover, while lexicon, phonology, and grammar are the three key aspects that distinguish each dialect from others, existing works in Dialectal NLP have so far mainly focused on the grammar variations of dialects (Ziems et al., 2022; 2023; Blodgett et al., 2018; Jørgensen et al., 2015). In this work, we provide the first large-scale dataset of dialectal lexical variations, bridging the gap towards holistic dialectal variation evaluation and building dialect-robust machine learning models.

## 3 DIALECTGEN BENCHMARK

### 3.1 DATASET CONSTRUCTION

To select dialect features for our benchmark dataset, we first gather dialect lexemes along with their dictionary definitions and example usages from publicly available regional English dictionaries including The Oxford Regional English Dictionary (Gates et al., 2023), Dictionary of American regional english (Cassidy et al., 1985), A dictionary of Singlish and Singapore English (Lee, 2004), Dictionary of Indian English (Subhash, 2020), and The Oxford Dictionary of African American English (Heinmiller, 2023). We collect a total of 1126 dialect lexemes for initial processing.

Based on the dictionary definitions of the selected lexemes, we manually filter out: (1) potentially derogatory lexemes; (2) culture-unique lexemes without Standard American English (SAE) equivalents. We then carefully read the dictionary definitions of each remaining dialect lexeme and assign it a SAE equivalent lexeme with the same meaning, creating a list of pair-wise corresponding lexical features for each dialect. Examples of selected pairs can be seen in Table 1 and Figure 1.

Next, we use GPT4o (Hurst et al., 2024) to generate prompts for each SAE word in our paired lexical feature set. We specifically instruct the model to generate prompts describing a visual scene with the lexeme playing a central role, which can be one of the following depending on the semantic role of the lexeme: (1) The central object in the scene; (2) The main action of the central object; (3) A prominent descriptive feature of the central object.

We also ask the model to create two different sets of **Concise** and **Detailed** prompts for each SAE lexeme. Then we simply replace the SAE lexeme in the prompts with the dialect lexeme (Table 1) to create our two dialect evaluation settings:

- **Concise** prompts generally consist of $\leq 6$ words, with the goal of providing a more challenging evaluation setting where the multimodal generative model is not given too many contextual hints about the lexeme's meaning.

- **Detailed** prompts generally consist of $\geq 9$ words, with the goal of providing a more relaxed evaluation setting where the multimodal generative model can use more contextual hints to infer the lexeme's meaning.

These two evaluation settings also intuitively represent two common user input styles for multimodal generative models, where casual users may tend to provide concise prompts and professional users may be more inclined to write detailed prompts. Across **Concise** and **Detailed** evaluation settings, we generate a total of 6552 prompts.

For specific Dialect Prompt / SAE Prompt pairs where the dialect lexeme has an additional polysemous meaning recorded in an SAE dictionary (Webster, 1869), we generate an additional SAE Polysemy Prompt, where the lexeme is used unambiguously in its SAE meaning. This data can be used for regulating model behavior in training scenarios.

## 3.2 DIALECT SPEAKER VALIDATION AND FILTERING

Before admitting the generated prompts to our final evaluation benchmark, we carefully verify their quality and correctness with dialect speaker human annotators. We created a specialized Amazon MTurk interface (Figure 4) for prompt annotation and matching potential dialect speaker annotators to their spoken dialect: each human annotator must first self-identify their dialect background and then complete a dialect speaker assessment quiz (Ziems et al., 2023) that matches each annotator to at most one dialect (Figure 5). Annotators are only selected if both their self-identified dialect background and their quiz assessment result match to the same dialect. More details on human annotation are available in Section E.

After each dialect speaker is selected, they will be presented with Dialect Prompt / SAE Prompt pairs where the only difference is the dialect lexeme being swapped with its SAE equivalent word. For each pair of prompts, the dialect speaker must answer two questions:

1. Does the given Dialect Prompt make sense in said Dialect and correspond exactly in meaning to the given SAE prompt in Standard American English? (Yes / No / I don't know)

2. Is the given Dialect Prompt ambiguous? *i.e.*, Does it have a reasonable alternative interpretation in the Standard American English (SAE) context? (Yes / No / I don't know)

Each Dialect Prompt / SAE Prompt pair is presented to two independent dialect speaker annotators. A pair is included in the final dataset only if both human annotators answer "Yes" to the first question and "No" to the second question. Consistent responses ensure the dialect prompt is: (1) exactly synonymous with the SAE prompt. (2) valid in the dialectal context. (3) non-ambiguous (for polysemous lexemes).

In total, dialect speaker filtering further removes 35.9% of all generated prompts, resulting in a final dataset containing 4,200 validated prompts.

## 4 EXPERIMENTS

### 4.1 EVALUATION METRICS

**Automatic Evaluation**  To automatically evaluate any multimodal generative model $\mathcal{G}(\cdot)$ on our benchmark, we design scoring functions based on reference-free image-text alignment metrics, including VQAScore (Lin et al., 2024) and CLIPScore (Hessel et al., 2021). For simplicity, we denote any such alignment metric below as $\mathcal{A}$. We further denote the **DialectGen** prompt subset for any dialect as $\mathcal{P}$, which contains many SAE Prompt / Dialect Prompt pairs $p = (p^s, p^d)$.

For each individual text prompt $p^s$ or $p^d$, we generate $n$ images under different random seeds for text-to-image generative models, or uniformly sample $n$ frames in a video for text-to-video generative models. Therefore, for each SAE Prompt / Dialect Prompt pair $p = (p^s, p^d) \in \mathcal{P}$, we can calculate its SAE and Dialect performance as follows:

$$SAE(p, \mathcal{G}) = \frac{1}{n} \sum_{i=1}^{n} \mathcal{A}(p^s, \mathcal{G}(p^s)_i) \qquad (1)$$

$$Dialect(p, \mathcal{G}) = \frac{1}{n} \sum_{i=1}^{n} \mathcal{A}(p^s, \mathcal{G}(p^d)_i) \qquad (2)$$

Table 2: **DialectGen** benchmark results for popular text-to-image and text-to-video generative models, including **Dialect-wise Performance Drop** measured by VQAScore (Lin et al., 2024); and **Overall Performance Drop** measured by human eval, VQAScore, and CLIPScore (Hessel et al., 2021). Cells are highlighted based on numerical value normalized across the entire table, with darker red indicating a higher performance drop in the given metric.

| | Model | Overall Performance Drop (%) ↓ | | | Dialect-wise Performance Drop (%) ↓ | | | | |
|---|---|---|---|---|---|---|---|---|---|
| | | Human | VQAScore | CLIPScore | AAE | BrE | ChE | InE | SgE |
| **Concise Prompts** — *T2I Models* | Stable Diffusion 1.4 | 28.19 | 26.7 | 10.35 | 20.67 | 9.64 | 34.94 | 41.27 | 26.96 |
| | Stable Diffusion 1.5 | 29.77 | 27.06 | 10.32 | 19.51 | 8.66 | 36.5 | 42.15 | 28.48 |
| | Stable Diffusion 2.1 | 31.46 | 28.79 | 11.7 | 24.35 | 9.31 | 44.82 | 41.12 | 28.89 |
| | Stable Diffusion XL | 29.8 | 26.69 | 10.88 | 23.37 | 7.95 | 41.22 | 38.74 | 22.17 |
| | Stable Diffusion 3 | 31.89 | 29.01 | 10.81 | 27.89 | 8.64 | 42.67 | 40.69 | 25.12 |
| | Stable Diffusion 3.5 Large | 32.31 | 29.43 | 11.37 | 28.3 | 9.74 | 42.66 | 41.9 | 24.56 |
| | Stable Diffusion 3.5 Large Turbo | 32.92 | 30.28 | 11.34 | 30.33 | 9.27 | 43.6 | 42.49 | 25.72 |
| | Flux.1 [dev] | 36.43 | 32.26 | 10.88 | 30.61 | 10.83 | 44.64 | 42.59 | 32.62 |
| | DALL-E Mini | 34.29 | 31.52 | 11.71 | 33.91 | 8.18 | 47.11 | 42.85 | 25.51 |
| | DALL-E 2 | 38.63 | 32.79 | 9.97 | 35.87 | 7.95 | 48.78 | 47.21 | 24.14 |
| | DALL-E 3 | 26.55 | 24.39 | 9.32 | 18.97 | 3.58 | 41.95 | 31.9 | 25.56 |
| | DALL-E 3 (w/ Prompt Rewrite) | 20.19 | 18.25 | 6.69 | 22.11 | 6.48 | 26.86 | 23.05 | 12.74 |
| | gpt-image-1 (4o Image Gen) | 22.18 | 19.18 | 7.65 | 26.12 | 5.2 | 26.09 | 26.51 | 11.99 |
| *T2V Models* | Cosmos-1 | 25.41 | 20.49 | 6.66 | 22.15 | 9.69 | 26.1 | 27.44 | 17.09 |
| | Open-Sora | 29.98 | 26.63 | 8.93 | 22.59 | 9.19 | 43.09 | 31.74 | 26.53 |
| | VideoCrafter-2 | 32.5 | 30.24 | 10.51 | 25.36 | 9.43 | 50.36 | 39.95 | 26.08 |
| | CogVideoX | 40.06 | 42.55 | 11.04 | 38.33 | 23.75 | 55.18 | 26.1 | 27.44 |
| | Wan 2.1 | 48.17 | 47.33 | 13.1 | 52.68 | 31.27 | 43.83 | 53.38 | 55.47 |
| **Detailed Prompts** — *T2I Models* | Stable Diffusion 1.4 | 14.33 | 15.93 | 5.16 | 11.65 | 4.37 | 17.35 | 29.23 | 17.03 |
| | Stable Diffusion 1.5 | 16.56 | 16.17 | 5.51 | 11.18 | 5.39 | 17.34 | 28.7 | 18.22 |
| | Stable Diffusion 2.1 | 17.39 | 18.4 | 5.78 | 15.06 | 5.51 | 23.03 | 29.36 | 19.06 |
| | Stable Diffusion XL | 17.12 | 17.09 | 5.83 | 14.09 | 5.56 | 20.57 | 30.12 | 15.1 |
| | Stable Diffusion 3 | 17.15 | 18.64 | 5.86 | 14.74 | 6.67 | 23.85 | 28.94 | 19.02 |
| | Stable Diffusion 3.5 Large | 18.42 | 19.54 | 6.12 | 15.7 | 6.99 | 23.46 | 31.83 | 19.72 |
| | Stable Diffusion 3.5 Large Turbo | 19.9 | 20.63 | 6.09 | 15.06 | 8.13 | 24.94 | 33.42 | 21.61 |
| | Flux.1 [dev] | 23.29 | 21.25 | 5.46 | 14.84 | 9.11 | 25.69 | 31.4 | 25.23 |
| | DALL-E Mini | 24.71 | 21.44 | 7.05 | 27.56 | 5.29 | 27.35 | 31.47 | 15.53 |
| | DALL-E 2 | 17.73 | 20.2 | 5.98 | 18.43 | 6.52 | 25.5 | 32.8 | 17.76 |
| | DALL-E 3 | 12.18 | 13.27 | 4.29 | 8.85 | 4.74 | 20.98 | 18.91 | 12.85 |
| | DALL-E 3 (w/ Prompt Rewrite) | 6.55 | 10.77 | 2.97 | 11.93 | 5.28 | 10.62 | 17.09 | 8.94 |
| | gpt-image-1 (4o Image Gen) | 8.98 | 10.97 | 3.24 | 13.72 | 4.46 | 10.56 | 15.96 | 10.17 |
| *T2V Models* | Cosmos-1 | 18.04 | 14.28 | 4.3 | 11.05 | 9.25 | 14.04 | 22.49 | 14.58 |
| | Open-Sora | 17.16 | 14.1 | 4.57 | 13.49 | 5.13 | 19.4 | 19.8 | 12.69 |
| | VideoCrafter-2 | 22.59 | 18.31 | 5.91 | 16.97 | 4.18 | 24.16 | 27.63 | 18.61 |
| | CogVideoX | 31.87 | 29.6 | 8.08 | 21.33 | 14.63 | 32.74 | 42.88 | 36.4 |
| | Wan 2.1 | 32.69 | 31.94 | 8.59 | 30.23 | 14.97 | 42.58 | 36.21 | 35.71 |

Note that when calculating dialect performance, we align the SAE Prompt $p^s$ with multimodal output generated from the corresponding Dialect Prompt, *i.e.*, $\mathcal{G}(p^d)$. This is feasible given that the paired prompts are synonymous, as verified by dialect speaker annotators in Section 3.2. Based on this, we can compute the dialect-induced performance drop of $\mathcal{G}(\cdot)$ for each prompt pair $p$:

$$Drop(p,\ \mathcal{G}) = \frac{SAE(p,\ \mathcal{G}) - Dialect(p,\ \mathcal{G})}{SAE(p,\ \mathcal{G})} = \sum_{i=1}^{n} \frac{\mathcal{A}(\mathcal{G}(p^s)_i,\ p^s) - \mathcal{A}(\mathcal{G}(p^d)_i,\ p^s)}{\mathcal{A}(\mathcal{G}(p^s)_i,\ p^s)} \quad (3)$$

To obtain the average model performance drop for a specific dialect, *i.e.*, $Drop(\mathcal{P},\ \mathcal{G})$, we simply average $Drop(p,\ \mathcal{G})$ for all $p$ in $\mathcal{P}$.

**Human Evaluation**    We further design a human evaluation pipeline to check the empirical alignment between our automatic evaluation metrics and human judgment. For 5% of the model outputs in our benchmark, we ask three independent external human annotators to evaluate: to what extent does the multimodal generations conditioned on the SAE Prompt $\mathcal{G}(p^s)$ or Dialect Prompt $\mathcal{G}(p^d)$ match with the scene described by SAE prompt $p^s$. Annotators are asked to rate the alignment between each (image/video, caption) pair with a numerical score between 0 and 10. The numerical scores are scaled by 0.1 to match the scoring range of VQAScore and CLIPScore before calculating SAE and Dialect performance. Finally, we use the same formula to calculate the dialect-induced performance drop $Drop(p,\ \mathcal{G})$. Since we only evaluate the alignment between image/video and the SAE prompt, this task does not require dialect speaker human annotators.

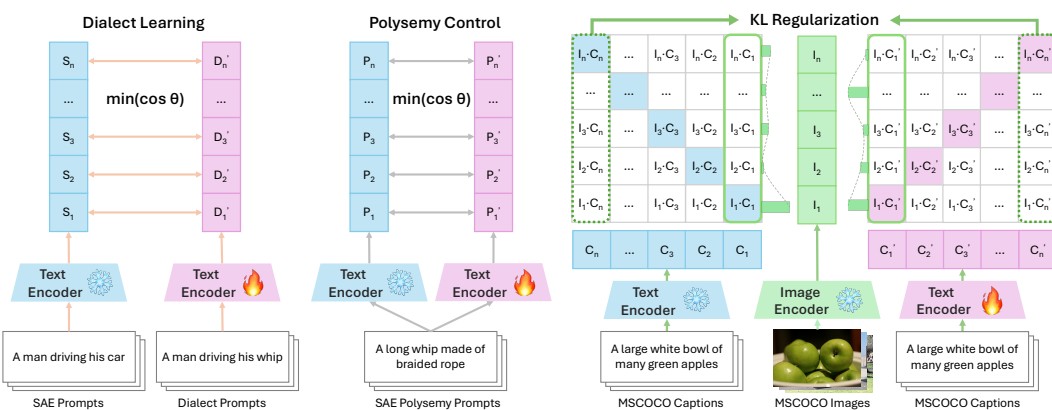

Figure 2: Losses used in our mitigation. Text prompts for **Dialect Learning** and **Polysemy Control** come from the **DialectGen** training set, while image-caption pairs for **KL Regularization** come from the MSCOCO validation set.

## 4.2 BENCHMARK EXPERIMENTS

Applying the automatic and human evaluation metrics described in Section 4.1, we evaluate popular open-weights and proprietary multimodal generative models on **DialectGen**. Model performances are separately aggregated for **Concise Prompts** and **Detailed Prompts** settings in Table 2.

**Overall Performances**    For each model, we record overall dialect-induced performance drop on **DialectGen** using three different metrics: Human Eval, VQAScore, and CLIPScore. We calculate Pearson correlation coefficients (Pearson, 1895) $r$ between each of the two metrics and observe $r$(Human, VQAScore) = 0.968, $r$(Human, CLIPScore) = 0.924, and $r$(VQAScore, CLIPScore) = 0.907. This shows that while both automatic scoring metrics have high correlations to human judgement (the gold standard), VQAScore is a better-aligned scoring metric for measuring dialect-induced performance drop.

Contrasting the model performance drops across the two evaluation settings **Concise Prompts** and **Detailed Prompts**, we can clearly see that all models exhibit significantly larger performance drops for concise prompts compared to detailed prompts. This is in line with our assumption that models can more easily infer the meanings of unknown dialect lexemes from richer prompt contexts, highlighting the need for challenging evaluation via concise prompts to reveal model robustness issues.

Looking at individual model performances, we observe that among text-to-video generative models: Wan 2.1 (Wang et al., 2025) and CogVideoX (Yang et al., 2024) exhibit the largest overall performance drops while Cosmos-1 (Agarwal et al., 2025) is the most robust. While for text-to-image generative models, DALL-E 2 (Ramesh et al., 2022) and Flux.1 [dev] (Black Forest Labs, 2024) exhibit the largest overall performance drops while DALL-E 3 (Betker et al., 2023) (w/ Prompt Rewrite) and gpt-image-1 (4o Image Generation) (OpenAI, 2025) are the most robust.

**Dialect-wise Performance Drop**    In addition to overall performance, we record each model's performance drop on each dialect, measured by VQAScore. Based on the color heatmap in Table 2, we can clearly see that the most severe performance drops occur for ChE and InE for most models, while AAE and SgE also suffer significant performance decreases. On the other hand, models generally do not see a very significant performance drop for BrE, which is expected given the relatively higher-resource nature of the dialect.

## 5 MITIGATION METHODS

The significant dialect performance drops of current multimodal generative models shown in Section 4.2 highlight the need for effective mitigation strategies to improve dialect robustness. Here, the goal is to develop a method that enhances robustness across multiple dialects while preserving

performance on standard SAE prompts. To this end, we first investigate intuitive baseline approaches, including (1) UNet Finetuning; (2) Prompt Revision, and then introduce our new mitigation strategy.

## 5.1 BASELINE METHODS

**UNet Finetuning** The vast majority of current text-to-image and text-to-video generative models comprise two main components: a text encoder and a diffusion-based image/video decoder. In current post-training paradigms, typically the text encoder is kept frozen while the diffusion UNet is fine-tuned (Podell et al., 2023; Rombach et al., 2022; Betker et al., 2023; Dai et al., 2023). Existing works in aligning, enhancing, and customizing multimodal generative models also focus heavily on developing reward-based fine-tuning methods for the diffusion UNet while freezing the text encoder (Segalis et al., 2023; Clark et al., 2023; Prabhudesai et al., 2023; Black et al., 2023; Fan et al., 2023; Wallace et al., 2024; Dang et al., 2025).

Based on existing works, we apply prominent multimodal generation enhancement methods towards improving dialect robustness, including:

- **Diffusion Fine-tune** (Rombach et al., 2022) given a pair of synonymous Dialect / SAE Prompts, we fine-tune the diffusion UNet with the Dialect Prompt as input, and images generated using the SAE Prompt as target output.
- **Diffusion DPO** (Wallace et al., 2024) We similarly use the Dialect Prompt as input, and use images generated with the SAE Prompt / Dialect Prompt as Win / Lose pairs for DPO.

**Prompt Revision** Beyond UNet fine-tuning, another popular family of methods for aligning and enhancing multimodal generative models is prompt revision (Hao et al., 2023; Betker et al., 2023; Wang et al., 2024; Chen et al., 2024). In our experiments, we include both a general prompt rewriting method and targeted prompt translation methods using general-purpose LLMs:

- **Prompt Rewrite** We apply the general prompt rewriting pipeline in Betker et al. (2023) to all test prompts before passing them to the generative model.
- **Prompt Translate** We use general-purpose LLMs (Grattafiori et al., 2024; OpenAI, 2025) to translate all prompts to SAE before passing them to the generative model.

## 5.2 OUR METHOD

Unlike prior approaches, we propose a new mitigation strategy that focuses on updating the text encoder(s). A natural first step toward improving dialectal robustness is to align the semantic representation of a dialect expression with that of its corresponding SAE counterpart.

**Dialect Learning** To operationalize this idea, we introduce a Dialect Learning loss that encourages the target text encoder to recognize dialectal lexemes by minimizing the cosine distance between the target encoder's embedding of a dialect prompt and the frozen encoder's embedding of its synonymous SAE prompt:

$$\mathcal{L}_{\text{DL}} = \frac{1}{N} \sum_{i=1}^{N} \left( 1 - \langle \pi(p_i^d),\ \pi_0(p_i^s) \rangle \right). \tag{4}$$

Here, $\langle \cdot,\ \cdot \rangle$ denotes cosine similarity; $\pi(\cdot)$ and $\pi_0(\cdot)$ represent the trainable target text encoder and the frozen reference encoder, respectively; and $p_i^d$ and $p_i^s$ denote the $i$-th pair of synonymous prompts in dialect and standard English, respectively. Although this may improve dialectal robustness, relying on this loss alone may compromise the model's ability to handle dialect lexemes that exhibit polysemy in SAE contexts.

**Polysemy Control** In order to retain the model's ability to correctly recognize polysemous lexemes within SAE contexts, we introduce a Polysemy Control loss that minimizes the cosine distance between embeddings of the same SAE polysemous prompt generated by the target and frozen encoders:

$$\mathcal{L}_{\text{PC}} = \frac{1}{N} \sum_{i=1}^{N} \left( 1 - \langle \pi(p_i^m),\ \pi_0(p_i^m) \rangle \right), \tag{5}$$

where each $p_i^m$ is a polysemous SAE prompt sampled from the dataset. This loss is applied only to examples containing SAE polysemous lexemes.

Table 3: **Mitigation results** for all baseline methods and our best performing method, including **Overall Performances** on SAE MSCOCO, SAE Polysemy, average Dialect performance, and **Dialect Performance** for each dialect, all measured using VQAScore Lin et al. (2024). Cell colors reflect column-normalized performance values, with darker green indicating higher VQAScore performance.

| Mitigation Methods | Overall Performances ↑ | | | Dialect Performance ↑ | | | | |
|---|---|---|---|---|---|---|---|---|
| | SAE MSCOCO | SAE Polysemy | Dialect Avg. | AAE | BrE | ChE | InE | SgE |
| **Base Model (Stable Diffusion 1.5)** | 75.49 | 72.84 | 57.80 | 60.13 | 69.39 | 52.65 | 49.94 | 56.89 |
| **Prompt Revision** | | | | | | | | |
| DALL-E 3 Prompt Rewrite | 74.25 | 70.85 | 60.91 | 57.34 | 69.51 | 56.36 | 57.54 | 63.81 |
| LLaMA 3 Prompt Translate | 74.03 | 71.33 | 58.48 | 57.73 | 70.4 | 53.98 | 50.42 | 59.87 |
| GPT4.1 Prompt Translate | 74.54 | 71.47 | 63.90 | 60.87 | 74.39 | 59.05 | 60.20 | 64.98 |
| **UNet Fine-tuning** | | | | | | | | |
| Diffusion Finetune | 65.01 | 52.13 | 60.94 | 63.85 | 70.14 | 57.3 | 52.84 | 60.56 |
| Diffusion DPO | 63.94 | 50.32 | 63.52 | 66.31 | 68.91 | 61.22 | 56.38 | 64.79 |
| **Our Encoder Tuning Methods** | | | | | | | | |
| Dialect Learning | 67.14 | 46.30 | 78.02 | 75.21 | 78.33 | 79.31 | 78.10 | 79.15 |
| + Text Cosine Reg. | 67.06 | 46.39 | 77.93 | 75.44 | 77.84 | 79.31 | 78.22 | 78.86 |
| + Image Cosine Reg. | 67.73 | 46.48 | 78.00 | 74.91 | 78.20 | 79.45 | 78.33 | 79.11 |
| + Text KL Reg. | 72.68 | 52.72 | 77.78 | 74.40 | 78.27 | 78.36 | 78.17 | 79.71 |
| + Image KL Reg. | 71.69 | 53.41 | 78.12 | 73.77 | 77.23 | 79.06 | 79.25 | 81.29 |
| + Text KL Reg. + Polysemy Ctrl. | 72.71 | 70.15 | 77.74 | 72.24 | 75.76 | 78.95 | 80.67 | 81.07 |
| + Image KL Reg. + Polysemy Ctrl. | 74.80 | 71.17 | 77.68 | 72.61 | 76.74 | 77.51 | 80.41 | 81.14 |

**KL Regularization** In addition to the previous two losses, it is also essential to preserve the model's performance on general SAE prompts. To this end, one might consider employing the conventional Kullback-Leibler (KL) divergence loss, which promotes alignment between the output distributions of a trainable target model and a frozen reference model over a predefined discrete logit space. However, this approach is not directly applicable in our setting, as text encoders output continuous embeddings rather than discrete logits. To address this challenge, we approximate the output distribution by computing similarity scores between a given caption embedding and a set of reference image embeddings drawn from a joint image-text embedding space. Concretely, we begin by sampling $M$ caption-image pairs $\left\{ (x_i^{\text{cap}}, x_i^{\text{img}}) \mid i \in [M] \right\}$ from a general SAE dataset such as MSCOCO (Lin et al., 2014). For each pair, we compute the caption embedding $C_i = \pi_0(x_i^{\text{cap}})$ using a frozen text encoder $\pi_0$, and the corresponding image embedding $I_i = \phi_0(x_i^{\text{img}})$ using a frozen image encoder $\phi_0$, with both encoders operating in the same shared text-image embedding space. The resulting image embeddings $\{I_i \mid i \in [M]\}$ serve as reference anchors for computing similarity scores with a given caption embedding. These scores act as surrogate logits that approximate the output distributions required for the KL divergence computation. Specifically, for each caption $x_i^{\text{cap}}$, we define the approximated output distributions for the frozen encoder $\pi_0$ and the trainable target encoder $\pi$ as:

$$
\begin{aligned}
\mathbf{s}_i^{\pi_0} &= [\langle I_1, C_i \rangle, \ldots, \langle I_M, C_i \rangle], \\
\mathbf{s}_i^{\pi} &= [\langle I_1, C_i' \rangle, \ldots, \langle I_M, C_i' \rangle],
\end{aligned}
\tag{6}
$$

where $C_i' = \pi(x_i^{\text{cap}})$.

Given these simulated logits, we define the KL divergence loss to encourage the target encoder's output distribution to remain close to that of the frozen encoder:

$$
\mathcal{L}_{\text{KL}} = \frac{1}{M} \sum_{i=1}^{M} \text{KL} \left( \text{softmax}(\mathbf{s}_i^{\pi}) \,\|\, \text{softmax}(\mathbf{s}_i^{\pi_0}) \right).
\tag{7}
$$

This approach is compatible with CLIP-style models (Radford et al., 2021; Zhai et al., 2023), in which image and text embeddings are aligned within a shared representation space. When an image encoder is unavailable, we instead use the frozen caption embeddings $\{C_i \mid i \in [M]\}$ as proxies for reference anchors. We hereafter refer to the case where image embeddings are used as reference anchors as "Image KL Reg." and the one using text embeddings as "Text KL Reg."

Based on these design choices, the final combined loss function integrates all three components: $\mathcal{L} = \mathcal{L}_{\text{DL}} + \mathcal{L}_{\text{PC}} + \mathcal{L}_{\text{KL}}$ as illustrated in Figure 2. For more details, please refer to Section B.

## 5.3 MITIGATION RESULTS

Here, we validate all baselines and our method on SD1.5 and SDXL. Due to space limitations, the results for SDXL are reported in Section F.

### 5.3.1 COMPARISON WITH THE BASELINES

As shown in Table 3, prompt rewriting methods that operate solely at the input level do not degrade SAE MSCOCO or polysemy performance, but yield only slight improvements up to 6.1% in average dialect performance. Furthermore, UNet fine-tuning approaches also lead to small gains of up to 5.7% in dialect performance, but at the cost of substantial drops in both general SAE and polysemy scores. In contrast, our method, corresponding to the last row of the table and incorporating all three loss components described in Section 5.2, significantly improves dialect robustness across all five dialects. Its average dialect performance of 77.68% closely approaches the base model's SAE score of 77.91%, while causing negligible degradation in SAE MSCOCO and polysemy performance.

### 5.3.2 ABLATION STUDY

To evaluate the contribution of each component in our method, we conduct an ablation study here.

**Base Model vs. Dialect Learning**   As shown in Table 3, applying the Dialect Learning loss ($\mathcal{L}_{\text{DL}}$) alone yields huge improvements in the base model's dialect performance, but also degrades SAE MSCOCO and polysemy performance.

**Cosine Reg. vs. KL Reg.**   To solve this issue, simply maximizing cosine similarity between the target text encoder's text embeddings and the corresponding text/image embeddings from the frozen text/image encoder (denoted as Text/Image Cosine Reg.), which is computed over the same caption-image pairs used in our KL regularization, does not effectively recover the base model's SAE MSCOCO and polysemy performance. In contrast, adding our KL regularization loss ($\mathcal{L}_{\text{KL}}$) improves both metrics while preserving dialect gains.

**Adding Polysemy Ctrl.**   Finally, incorporating the Polysemy Control loss ($\mathcal{L}_{\text{PC}}$) yields substantial gains in polysemy performance, improving it by 17.43% and 17.76% for Text and Image KL Reg. respectively, underscoring the importance of this component in recognizing polysemous lexemes within SAE contexts.

## 6 LIMITATIONS

Our study focuses on the lexical variations that characterize dialects, motivated by the empirical observation that such variations exert much greater influence on multimodal generative model performance than grammatical variations (see Section H). Furthermore, grammatical variation has already been the subject of extensive investigation in text-only contexts (Hudson, 1996; Chambers & Trudgill, 1998; Fromkin et al., 1998; Nerbonne, 2009; Wardhaugh & Fuller, 2021). These considerations jointly motivate our decision to prioritize the evaluation of lexical dialect variation, which appears especially consequential in the multimodal generative setting. Furthermore, our evaluation of text-image alignment utilizes reference-free metrics, namely VQAScore (Lin et al., 2024) and CLIPScore (Hessel et al., 2021). We recognize that these pretrained vision-language models are not perfect. To address this potential weakness, we conducted a thorough human evaluation and found very high statistical correlation between our automatic metrics and human judgment (Pearson correlation coefficient $r = 0.968$ for VQAScore and $r = 0.924$ for CLIPScore). Therefore, while acknowledging the imperfections of automated metrics, this high degree of human correlation provides strong evidence for the validity of our evaluation metrics and associated analysis conclusions.

## 7 CONCLUSIONS

In this work, we create **DialectGen**, a large-scale multi-dialectal benchmark evaluating the dialect robustness of multimodal generative models. Our experiments on 17 widely used text-to-image and text-to-video generative models reveal severe performance drops up to 38.63% and 48.17% for image and video generative models, respectively. We further design an encoder-based mitigation strategy to enhance dialect robustness while preserving performance on Standard American English.

## 8 ETHICS STATEMENT

This work makes use of human subjects for annotation and evaluation. All procedures were subject to ethical review and were approved by the IRB from the authors' institution. Consent was gathered in accordance with the authors' institution guidelines, and annotators had access to a data use statement when giving consent. The purpose of **DialectGen** is to provide tools that enable researchers and practitioners to evaluate and improve dialect robustness in their models. We will release these data responsibly, ensuring that users sign a Data Use Agreement that forbids the use of **DialectGen** for deception, impersonation, mockery, discrimination, hate speech, targeted harassment, and cultural appropriation. In the agreement, researchers and practitioners will also acknowledge the limitations of this work, that **DialectGen** may not fully or accurately represent the natural usage patterns of all sub-communities of speakers. **DialectGen** is designed to be easily updatable and configurable, such that it can be extended by and for specific sub-communities and updated as dialects evolve over time. We have carefully checked our data to make sure no personally identifying information or offensive content is included. When utilizing existing artifacts and models, we make sure to follow all relevant regulations and licenses.

## 9 REPRODUCIBILITY STATEMENT

**Reproducibility Statement.** We have taken several steps to ensure the reproducibility of our work. Detailed descriptions of dataset construction, annotation procedures, evaluation protocols, and mitigation methods are provided in the main paper (see Sections 3, 4, etc.), with further implementation details, training configurations, and additional qualitative results included in the appendix (see Sections B, A, etc.). To facilitate independent verification, we also provide as anonymized supplementary material both the DialectGen benchmark dataset and the source code used for data processing, model training, and evaluation. The dataset files include all validated dialect–SAE prompt pairs, while the code folder contains scripts for dataset generation, automatic and human evaluation, and reproduction of all tables and figures reported in the paper. Together, these resources enable researchers to replicate our experimental results and extend the benchmark for future work.

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

# APPENDIX

## A QUALITATIVE COMPARISON

In Figure 3, we provide additional qualitative examples to demonstrate the performances of the baseline mitigation strategy, Diffusion DPO (Wallace et al., 2024), compared with our method. Specifically, we update the Stable Diffusion 1.5 model encoder using Dialect Learning, Polysemy Control, and Image KL. After mitigation, we ask each model to generate images based on the four dialect prompts first mentioned in Figure 1. The Stable Diffusion 1.5 Base model struggles to generate correct images for most of these prompts, including "Two ang pows on a table", "A man selling brinjal", and "A man hiking with his carnal". While the model is able to generate moderately reasonable images for the prompt "A man driving his whip", it commonly generates physically implausible details such as the man's torso protruding through the car. Fine-tuning the UNet with Diffusion DPO is able to slightly improve generation alignment with the text prompt (*e.g.*, occasionally generating two people for the prompt "A man hiking with his carnal"). However, it more often blends visual elements within the desired target images with other irrelevant objects (*e.g.*, generating a man selling purple pastries in place of eggplants or a man wearing a purple shirt holding vegetables). Our method generates higher-quality and better aligned images compared to the base model and Diffusion DPO by accurately learning to generate the target concepts without negatively impacting image quality. A significant majority of images in our sampled generations are able to generate images that correctly depict the target prompts, in line with quantitative evaluation results.

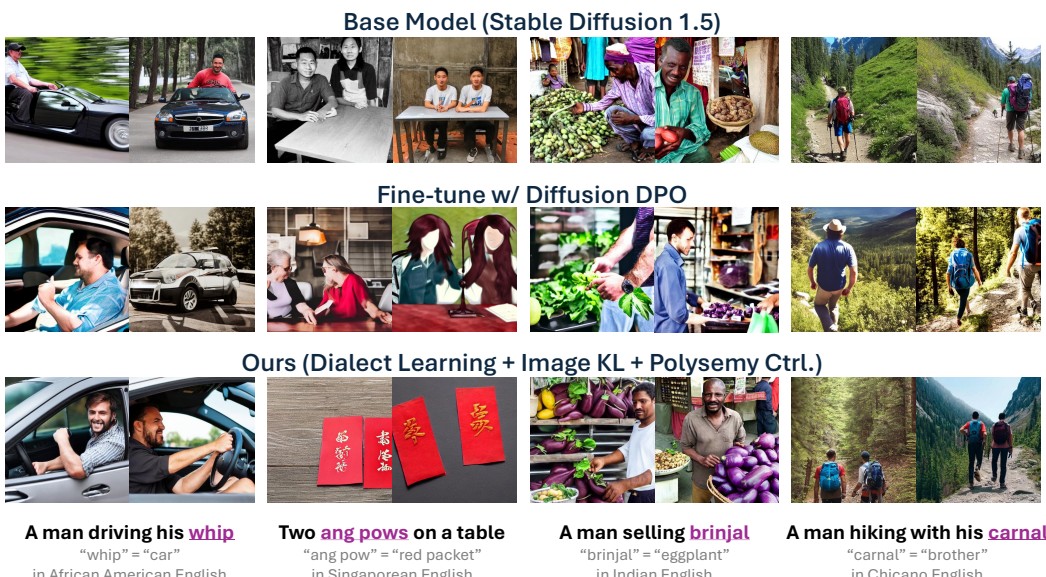

Figure 3: **Qualitative Comparison of Mitigation Strategies** using the Stable Diffusion 1.5 model (Rombach et al., 2022) on four different dialect prompts. Specifically, we compare the dialect prompt image generation results of the Stable Diffusion 1.5 Base Model, Stable Diffusion 1.5 fine-tuned with Diffusion DPO (Wallace et al., 2024), and Stable Diffusion 1.5 updated via our best performing method (Dialect Learning + Image KL Regularization + Polysemy Control).

## B IMPLEMENTATION DETAILS

**Data Preparation**  We first split the **DialectGen** dataset into training, validation, and test sets in a ratio of 80%, 10%, and 10%, respectively. These training and validation splits of **DialectGen** are used to compute the Dialect Learning loss and the Polysemy Control loss. For KL Regularization loss, we randomly sample 1,024 and 256 image-caption pairs from the MSCOCO validation set (Lin et al., 2014) for use in training and validation, respectively. The target text encoder is evaluated on

the validation set at the end of each epoch, and the checkpoint with the lowest validation loss is selected and saved for final evaluation. We then evaluate SAE polysemy and per-dialect performance using the test split of **DialectGen**, and assess SAE MSCOCO performance on 50 randomly sampled captions from the MSCOCO validation set.

**Training**   We employ the pretrained text encoder and fine-tune it for 30 epochs using the AdamW optimizer with an initial learning rate of $1 \times 10^{-4}$, $\beta_1 = 0.9$, $\beta_2 = 0.999$, and $\epsilon = 1 \times 10^{-8}$. A cosine annealing learning rate scheduler is applied across the 30 training epochs. The batch size, *i.e.*, $N$ in Equation (4) and Equation (5), is set to 32, and the number of image-caption pairs used for KL regularization, *i.e.*, $M$ in Equation (7), is set to 1,024. Training is completed in less than one hour on a single NVIDIA RTX A6000 GPU. In the case of SDXL, which includes both Base and Refiner encoders, the number of pairs $M$ for the Refiner encoder is set to 512 due to its larger size, and training takes approximately one hour using four NVIDIA RTX A6000 GPUs, with all other configurations kept the same as in the Stable Diffusion 1.5 and SDXL Base encoder settings.

**About T2Video Models**   Video-generation models incur substantially higher computational cost than their image counterparts. Since our primary goal is to assess the models' ability to interpret and render textual prompts, we generate only a small, fixed number of frames per video. This strategy is justified by two observations: (i) the first few frames typically suffice to judge prompt fidelity, and (ii) our prompts do not exhibit extensive motion, so long sequences offer diminishing returns.

All models were obtained by cloning their official repositories and following the authors' installation instructions. Frame numbers were uniformly reduced frame counts when possible, and in some cases, spatial resolution was also reduced to facilitate efficient evaluation—see Table 5 for the precise settings.

Average time per video was measured on a single NVIDIA RTX A6000 GPU; the Wan2.1-T2V-14B model, which does not fit in single-GPU memory, was benchmarked using six A6000 GPUs under Fully Sharded Data Parallel (FSDP) supported by the repository under the xdit framework.

All models except Wan2.1 fit under a single A6000 GPU and use approximately 20-30 GB of VRAM max. Wan2.1 takes at least 3 GPUs, taking an approximate memory usage of  100GB of combined VRAM.

## C   MODEL DETAILS

We provide detailed information on the multimodal generative models and key experimental settings used in our benchmark.

Table 4 lists the comprehensive specifications for all models evaluated in our work, including both text-to-image and text-to-video models. For each model, we provide details such as its creator organization, initial release date, hosting platform, availability type (*e.g.*, open source, proprietary), and model size.

Table 5 describes in detail the key generation parameters used for the text-to-video models. This includes the specific resolution, number of frames, and inference steps used for each model. Furthermore, we specify the average time required to generate a single video and the total time needed to generate our full video dataset to aid in understanding the reproducibility and computational cost of our experiments.

## D   DATASET DETAILS

The final **DialectGen** Dataset contains a total of 4632 prompts, which include 2100 non-SAE dialect prompts, 2100 SAE prompts, and 432 polysemous SAE prompts. The entire dataset is split into three subsets: training, validation, and test. The data split ratio is train : validation : test = 8 : 1 : 1. All benchmarking experiments are performed on the entire dataset, while for mitigation experiments, models are trained on the **DialectGen** training set while evaluated on the validation set.

Table 4: **Detailed Model Specifications** for all multimodal generative models (text-to-image and text-to-video generative models) benchmarked in this work. For reference and reproducibility, we include model name, model type, creator organization, initial release date, hosting platform, availability type, and model size.

| Model Name | Model Type | Created by | Release Date | Hosted by | Availability Type | Model Size |
|---|---|---|---|---|---|---|
| Stable Diffusion 1.4 | Text to Image | CompVis | 8/22/2022 | Hugging Face | Open Source | 1B |
| Stable Diffusion 1.5 | Text to Image | Runway ML | 10/20/2022 | Hugging Face | Open Weights | 1.3 B |
| Stable Diffusion 2.1 | Text to Image | Stability AI | 12/7/2022 | Hugging Face | Open Weights | 1.3 B |
| Stable Diffusion XL | Text to Image | Stability AI | 7/26/2023 | Hugging Face | Open Weights | 6.6 B |
| Stable Diffusion 3 Medium | Text to Image | Stability AI | 6/12/2024 | Hugging Face | Open Weights | 2 B |
| Stable Diffusion 3.5 Large | Text to Image | Stability AI | 10/22/2024 | Hugging Face | Open Weights | 8.1 B |
| Stable Diffusion 3.5 Large Turbo | Text to Image | Stability AI | 10/22/2024 | Hugging Face | Open Weights | 8.1 B |
| Flux.1 [dev] | Text to Image | Black Forest Labs | 4/2/2024 | Hugging Face | Open Weights | 12B |
| DALL-E Mini | Text to Image | Boris Dayma et al. | 7/25/2022 | Github | Open Weights | 0.4 B |
| DALL-E 2 | Text to Image | OpenAI | 9/28/2022 | OpenAI | Proprietary | N/A |
| DALL-E 3 | Text to Image | OpenAI | 8/20/2023 | OpenAI | Proprietary | N/A |
| gpt-image-1 | Text to Image | OpenAI | 4/23/2025 | OpenAI | Proprietary | N/A |
| VideoCrafter-2 | Text to Video | Tencent | 1/26/2024 | Hugging Face | Open Weights | 1.4 B |
| Open-Sora | Text to Video | HPC-AI Tech | 6/17/2024 | Hugging Face | Open Weights | 1.2 B |
| CogVideoX | Text to Video | THUDM Lab | 8/27/2024 | Hugging Face | Open Weights | 5 B |
| Cosmos-1 | Text to Video | Nvidia | 1/6/2025 | Hugging Face | Open Weights | 7 B |
| Wan 2.1 | Text to Video | Alibaba | 2/22/2025 | Hugging Face | Open Weights | 14 B |

Table 5: **Key Generation Parameters for Text-to-Video Generative Models**. For reproducibility and computational cost estimation, we list GPU runtime per video in minutes and GPU runtime for the full video dataset (both concise and detailed = 4110 videos) in hours. All computational costs are estimated for NVIDIA-A6000 GPUs with 48 GB Memory.

| Model Version | Resolution | Frames | Steps | Time / Video (min) | Time / Dataset (h) |
|---|---|---|---|---|---|
| VideoCrafter2 | $512 \times 512$ | 16 | 50 | 5.0 | 342.5 |
| OpenSora-STDiT-v3 | $405 \times 720$ | 51 | 30 | 8.3 | 570.8 |
| CogVideoX-5b | $720 \times 480$ | 10 | 10 | 6.1 | 416.7 |
| Cosmos-1.0-Diffusion-7B-Text2World | $704 \times 1280$ | 121 | 35 | 26.5 | 1815.3 |
| Wan2.1-T2V-14B | $832 \times 480$ | 10 | 12 | 4.8 | 329.4 |

*Note: The dataset-scale timing for Wan2.1-T2V-14B was measured using 6 A6000 GPUs using xdit FSDP.*

# E   HUMAN ANNOTATION DETAILS

≡ View Instructions

| Singlish (SgE) Prompt | Standard American English (SAE) Prompt |
|---|---|
| ${Dialect_Prompt} | ${SAE_Prompt} |

Q1: Does the above Singlish (SgE) Prompt make sense and correspond in meaning with the Standard American English (SAE) Prompt?

○ Yes   ○ No   ○ I don't know

Q2: Is the above Singlish (SgE) Prompt ambiguous? i.e. Does it have a reasonable alternative interpretation in the Standard American English (SAE) context?

○ Yes   ○ No   ○ I don't know

Submit

**Instructions**

The Dialectal Prompts and Standard American English Prompts below are constructed via examples from the Oxford English Dictionary and online crowdsourcing. Your role as an annotator is to further certify the prompts by answering the two questions below based on your knowledge, intuition, and any external resources you may find helpful.

**Example 1**

AAE Prompt:

A man driving his **whip**.

SAE Prompt:

A man driving his **car**.

In this case, the African American English (AAE) prompt is **not ambiguous** because although the word "whip" has a different meaning in Standard American English. The sentence structure (driving a whip) makes that interpretation unreasonable. Therefore, you should choose **"Yes"** for both Q1 and Q2 in this case.

**Example 2**

AAE Prompt:

A woman playing **football**.

SAE Prompt:

A woman playing **soccer**.

In this case, the British English (BE) prompt is **ambiguous** because it has a reasonable alternative interpretation in the Standard American English (SAE) context (the woman could be playing American football or gridiron instead of soccer). Therefore, you should choose **"Yes"** for Q1 and **"No"** for Q2 in this case.

Figure 4: **The Amazon Mechanical Turk Data Annotation Interface** for dialect speaker human filtering of generated prompts (prompt generation details in Section 3). Human annotators may use the "View Instructions" button to collapse / re-open detailed annotation instructions at any time. The annotation interface places no maximum time limit on each annotation question. Human annotators are allowed to return to previously annotated questions and update their answers at any time.

In the creation of the **DialectGen** Dataset, we recruit a total of 17 dialect speaker human annotators from Amazon Mechanical Turk. The demographic involves six annotators from Asia, eight annotators from North America, and 3 annotators from Europe. Each selected annotator is given the option to complete any number of questions as they prefer. We encourage each annotator to take regular breaks during the task and not to work consecutively for more than 2 hours on our task. Our task is relatively simple for dialect speakers as it mainly involves judging the plausibility and meaning of a sentence in their native dialect. We estimate each HIT to take around 12 seconds, this corresponds to an hourly wage of $15 USD. Our total annotation time is 21.84 hours, costing a total of $327.6. We ran 4 rounds of annotations, with a combined total of 6552 prompts. 35.9% of total proposed prompts were rejected by the annotators while 64.1% of prompts were approved.

# F   MITIGATION RESULTS ON STABLE DIFFUSION XL

Stable Diffusion XL consists of two encoders: a Base encoder and a Refiner encoder. We fine-tuned both components as part of our method. However, since the corresponding CLIP-style image encoder for the Refiner is not publicly accessible, only Text KL Regularization can be applied in this case. Given the Refiner's larger size and additional encoding modules, we evaluate our final method against other baselines within this more complex configuration.

## English Dialect Survey

**Instructions**  (Expand/Collapse)

**Purpose:** The purpose of this survey is to decide which of 77 cataloged varieties of English you might speak, in addition to the variety that is used in these instructions (SAE). No one variety is better or worse than any other, but each follows its own unique set of rules about what is *possible* (i.e. *well-formed*, *grammatical*, *acceptable*) to say.

**Task:** You see below a Dialect Excerpt (#2) from one of these 77 varieties along with its SAE Gloss (#1) above that. You should mark **Acceptable** if and only if the following are true:

1. (#2) follows all of the rules and constraints about what is possible to say in your variety of English.
2. (#1) and (#2) can both be used to mean the same (or nearly the same) thing, according to your understanding.

Otherwise, mark **Not Acceptable**.

**Note:** the following should **NOT** be used to decide if an excerpt is **Acceptable**:

1. Whether or not the excerpt is meaningful.
2. Whether or not you've seen the excerpt before.
3. Whether or not you think it is statistically probable that someone *else* might say it.
4. Whether or not you think the excerpt is *socially* acceptable.

Rely only on you understanding of how language works in the particular variety of English you speak.

**Tip:** You may speak *multiple* different varieties of English, but this quiz will work best if you focus on just one. To single one out, maybe you can imagine you are speaking only with one particular group of people in one particular setting.

**Examples of Linguistic Acceptability**  (Expand/Collapse)

Here, we list some examples of linguistic acceptability judgments for SAE, taken from (Warstadt et al. 2019).

Word Order

- **Acceptable:** Bo read the book.
- **Not Acceptable:** Read Bo the book.
- **Not Acceptable:** Bo the book read.

Subject-Verb Agreement

- **Acceptable:** My friend has to go.
- **Not Acceptable:** My friend have to go.

Causative-Inchoative Alternation

- **Acceptable:** The bubble popped.
- **Not Acceptable:** The bubble blew.

**(1) SAE Gloss:**

> *It's only five miles away.*

**(2) Dialect Excerpt:**

> *It's only five **mile** away.*

Description: Absence of plural marking only after quantifiers

[ Acceptable ]   [ Not Acceptable ]

**Match Code:**

**Matched Dialect:**
(Not Decided Yet)

Figure 5: **The English Dialect Speaker Assessment Quiz** used for matching dialect speaker annotators to specific dialects for prompt annotation. We adapt the assessment quiz from the existing English Dialect Speaker Survey first created in MultiVALUE (Ziems et al., 2023), which asks the human annotator to select their linguistic acceptability preference for 10 different dialect excerpts.

We report the mitigation results on Stable Diffusion XL (Podell et al., 2023) in Table 6, under the experimental setup described above. Similar to the findings on Stable Diffusion 1.5, Prompt Revision methods preserve general SAE performance but yield only marginal improvements in dialect VQAScore, with gains of up to 7.8%. Additionally, UNet fine-tuning methods also result in small gains of up to 5.3% in dialect performance, but at the cost of noticeable degradation in both SAE MSCOCO and SAE polysemy performance. In contrast, our method substantially improves dialect robustness across all five dialects, achieving an average performance of 85.99%, which surpasses the

Table 6: **Mitigation results on SDXL (Podell et al., 2023)** for all methods, including **Overall Performances** on SAE MSCOCO, SAE Polysemy, average Dialect performance, and **Dialect Performance** for each dialect, all measured using VQAScore (Lin et al., 2024). Cell colors reflect column-normalized performance values, with darker green indicating higher VQAScore performance.

| Mitigation Methods | Overall Performances ↑ | | | Dialect Performance ↑ | | | | |
| --- | --- | --- | --- | --- | --- | --- | --- | --- |
| | SAE MSCOCO | SAE Polysemy | Dialect Avg. | AAE | BrE | ChE | InE | SgE |
| **Base Model (Stable Diffusion XL)** | 86.21 | 78.21 | 61.55 | 61.17 | 77.58 | 47.04 | 53.21 | 68.76 |
| **Prompt Revision** | | | | | | | | |
| DALL-E 3 Prompt Rewrite | 85.36 | 78.01 | 66.49 | 59.93 | 77.92 | 60.61 | 63.62 | 70.39 |
| LLaMA 3 Prompt Translate | 84.72 | 77.60 | 64.19 | 63.74 | 77.93 | 57.40 | 56.09 | 65.80 |
| GPT4.1 Prompt Translate | 85.93 | 78.12 | 69.30 | 61.97 | 82.24 | 63.87 | 65.45 | 72.97 |
| **UNet Fine-tuning** | | | | | | | | |
| Diffusion Finetune | 70.49 | 52.37 | 65.22 | 65.31 | 76.69 | 60.12 | 58.05 | 65.91 |
| Diffusion DPO | 72.03 | 50.29 | 66.89 | 65.97 | 78.12 | 62.88 | 60.10 | 67.40 |
| **Ours** | | | | | | | | |
| Dialect Learning + Text KL Reg.+ Polysemy Reg. | 85.45 | 78.08 | 85.99 | 82.43 | 84.71 | 85.97 | 89.70 | 87.14 |

base model's SAE score of 84.43%, while inducing less than a 1% drop in both SAE MSCOCO and SAE polysemy performance.

## G   USE OF AI TOOLS

We employed large language models (LLMs), including OpenAI's GPT-5 and GPT-4o, as auxiliary tools to refine the manuscript and identify grammatical errors. All LLM-assisted content was critically reviewed, fact-checked, and revised by the authors to ensure scientific validity and originality. The authors retain full responsibility for all statements and conclusions presented in this work. Specifically, LLMs were used only to improve wording and clarity of expression.

## H   GRAMMATICAL VS. LEXICAL ROBUSTNESS IN MULTIMODAL MODELS

To establish the rationale for our study's focus on lexical variations, we begin with an observation about multimodal generative models. These models often exhibit a notable insensitivity to grammatical or syntactic structure, a tendency that likely arises from the bag-of-words nature of their CLIP-style encoders. This architectural trait means that variations in sentence construction, such as word order or verb tenses, tend to have a minimal effect on the final output. Table 7, adapted from Multi-VALUE (Ziems et al., 2023), showcases several examples of these grammatical variations.

Table 7: **Examples of Grammatical Dialect Variations** between Standard American English (SAE) sentences and African American English (AAE) dialect sentences. The **blue** texts highlight unique features in SAE while the **purple** texts (if applicable) highlight corresponding features in AAE.

| Grammatical Variation Type | SAE Prompt | AAE Dialect Prompt |
| --- | --- | --- |
| Clause Structure | A chair **that** can be folded | A chair can be folded |
| Negative Concord | There **is** no food on the table | There **ain't** no food on the table |
| Word Order | A **big and fresh** fish | A fish **big and fresh** |
| Verb Morphology | Mom **brought** rice to me | Mom **brin** rice give me |

To formally quantify this observation, we conducted a small-scale experiment with three representative models in the African American English evaluation setting. We used the Multi-VALUE (Ziems et al., 2023) translation system to apply grammatical variations to 300 SAE prompts from **DialectGen** and evaluated their generation quality using VQAScore.

The results, presented in Table 8, provide strong quantitative evidence supporting our initial analysis. While **lexical feature variations cause significant performance drops** for existing text-to-image generative models, **grammatical variations do not incur significant performance drops.** This

clear distinction validates our decision to focus on the more impactful lexical variations throughout this work.

Table 8: **Quantitative Effects of Grammatical and Lexical Variations on Multimodal Generation**, measured in VQAScore. We evaluate three text-to-image generative models under the following dialectal variation types: Grammatical, Lexical, and Grammatical + Lexical. Values in parentheses indicate the percentage performance drop in VQAScore compared to baseline SAE performance.

| Model | SAE Performance (%) | Performance under Dialectal Variations (%) | | |
|---|---|---|---|---|
| | | Grammatical | Lexical | Grammatical + Lexical |
| DALL-E Mini | 75.63 | 74.72 (-1.20) | 51.92 (-31.35) | 51.26 (-32.22) |
| FLUX.1 dev | 82.94 | 82.40 (-0.65) | 61.88 (-25.39) | 61.02 (-26.43) |
| Stable Diffusion 3.5 Large | 85.18 | 83.91 (-1.49) | 65.37 (-23.26) | 63.80 (-25.10) |

## I    PERFORMANCE BY DIALECT

Due to space constraints, we report performance by dialect in Table 9, Table 10, and Table 11. As described in Section 4.1, the scoring functions are based on reference-free image-text alignment metrics, including VQAScore and CLIPScore. We denote the subset of **DialectGen** prompts corresponding to a given dialect as $\mathcal{P}$, which consists of multiple SAE Prompt / Dialect Prompt pairs $p = (p^s, p^d)$. For each individual text prompt $p^s$ or $p^d$, we generate $n$ images under different random seeds for text-to-image generative models, or uniformly sample $n$ frames for text-to-video generative models. Accordingly, for each SAE Prompt / Dialect Prompt pair $p = (p^s, p^d) \in \mathcal{P}$, we compute its SAE and Dialect performance using Equation (1) and Equation (2), respectively. More concretely, $SAE(p, \mathcal{G})$ in Equation (1) denotes the average VQAScore (as reported in Table 9 and Table 10) or CLIPScore (in Table 11) computed over the $n$ images generated from the SAE prompt $p^s$. Similarly, $Dialect(p, \mathcal{G})$ in Equation (2) is computed using the same evaluation pipeline, but with the corresponding dialect prompt $p^d$ from the same pair. Each value of $SAE(p, \mathcal{G})$ and $Dialect(p, \mathcal{G})$ is reported as **SAE** and **Dialect**, respectively, in the tables.

## J    FUTURE WORK

Our work highlights several promising directions for future research, which we encourage the community to explore.

**Investigating Cultural and Representational Biases**    It would be interesting for future works to explore and evaluate the significance of representational and skin tone shifts induced by dialect inputs. For instance, as noted in Figure 1, we observed that FLUX.1 [dev] (Black Forest Labs, 2024) image generations for the prompt "A man selling eggplant" depict more upscale and decorated environments compared to generations for "A man selling brinjal." Furthermore, individuals depicted in the images for "brinjal" are darker-skinned. A systematic study of these shifts would provide valuable insights into the inherent biases of large-scale multimodal models.

**Exploring Grammatical and Joint Dialect Variations**    While this work concentrated on lexical variations, we welcome future works in this line to carefully study the impacts of grammatical dialect variations and their joint effects with lexical variations. Such research could reveal more complex interactions and failure modes in the performance of multimodal generative models.

**Extending Evaluation to Multi-Lexeme Prompts**    Another related area for future work is the extension of our evaluation to settings where multiple dialect lexemes are used. This would test the models' compositional understanding of dialectal language, and we encourage future works to explore such possibilities. However, it should be noted that creating high-quality, controlled data at scale for such experiments is a non-trivial problem that needs to be addressed.

Table 9: **Stable Diffusion 1.5 Mitigation Performance Breakdown** by dialect for different mitigation methods on the **DialectGen** dataset for all baseline methods and ablations of our method. All performance scores are measured using VQAScore (Lin et al., 2024), higher score is better.

| Mitigation Methods | Performance by Dialect (VQAScore) ↑ | | | | | | | | | |
|---|---|---|---|---|---|---|---|---|---|---|
| | AAE | | BrE | | ChE | | InE | | SgE | |
| | Dialect | SAE | Dialect | SAE | Dialect | SAE | Dialect | SAE | Dialect | SAE |
| **Base Model (Stable Diffusion 1.5)** | 57.34 | 72.94 | 69.51 | 76.40 | 56.36 | 78.66 | 57.54 | 81.05 | 63.81 | 80.50 |
| **Prompt Revision** | | | | | | | | | | |
| DALL-E 3 Prompt Rewrite | 57.73 | 73.16 | 70.40 | 77.86 | 53.98 | 79.99 | 50.42 | 81.33 | 59.87 | 81.66 |
| LLaMA 3 Prompt Translate | 60.87 | 70.36 | 74.39 | 76.49 | 59.05 | 78.15 | 60.22 | 81.09 | 64.98 | 79.84 |
| GPT4.1 Prompt Translate | 65.32 | 71.28 | 73.52 | 76.40 | 58.32 | 78.29 | 53.04 | 81.03 | 65.12 | 79.83 |
| **UNet Fine-tuning** | | | | | | | | | | |
| Diffusion Finetune | 63.85 | 64.34 | 70.14 | 68.35 | 57.30 | 69.55 | 52.84 | 70.72 | 60.56 | 72.42 |
| Diffusion DPO | 66.31 | 63.02 | 68.91 | 69.17 | 61.22 | 67.83 | 56.38 | 70.94 | 64.79 | 71.85 |
| **Ours** | | | | | | | | | | |
| Dialect Learning | 75.21 | 74.31 | 78.33 | 78.34 | 79.31 | 80.20 | 78.10 | 79.90 | 79.15 | 78.33 |
| + Text Cosine Reg. | 75.44 | 74.86 | 77.84 | 77.52 | 79.31 | 79.74 | 78.22 | 80.13 | 78.86 | 79.21 |
| + Image Cosine Reg. | 74.91 | 74.83 | 78.20 | 78.22 | 79.45 | 80.32 | 78.00 | 80.00 | 79.11 | 78.72 |
| + Text KL Reg. | 74.40 | 73.97 | 78.27 | 79.40 | 78.36 | 80.72 | 78.17 | 78.24 | 79.71 | 78.66 |
| + Image KL Reg. | 73.77 | 74.36 | 77.23 | 77.60 | 79.06 | 80.43 | 79.25 | 80.99 | 81.29 | 79.54 |
| + Text KL Reg.+ Polysemy Ctrl. | 72.24 | 72.25 | 75.76 | 79.57 | 78.95 | 79.27 | 80.67 | 79.89 | 81.07 | 79.84 |
| + Image KL Reg.+ Polysemy Ctrl. | 72.61 | 74.30 | 76.74 | 76.77 | 77.51 | 78.83 | 80.41 | 80.85 | 81.14 | 78.15 |

Table 10: **Complete DialectGen Benchmark Performance Breakdown** by dialect for all text-to-image and text-to-video generative models. All performance scores are measured using VQAScore (Lin et al., 2024), higher score is better. Results complements Table 2 in the main paper.

| | Model | Performance by Dialect (VQAScore) ↑ | | | | | | | | | |
|---|---|---|---|---|---|---|---|---|---|---|---|
| | | AAE | | BrE | | ChE | | InE | | SgE | |
| | | Dialect | SAE | Dialect | SAE | Dialect | SAE | Dialect | SAE | Dialect | SAE |
| Concise Prompts — T2I Models | Stable Diffusion 1.4 | 60.66 | 76.47 | 71.46 | 79.08 | 51.31 | 78.86 | 47.5 | 80.88 | 57.64 | 78.92 |
| | Stable Diffusion 1.5 | 62.31 | 77.41 | 72.59 | 79.47 | 50.4 | 79.37 | 47.03 | 81.29 | 56.36 | 78.8 |
| | Stable Diffusion 2.1 | 60.97 | 80.59 | 76.37 | 84.21 | 45.88 | 83.15 | 50.63 | 85.99 | 58.53 | 82.31 |
| | Stable Diffusion XL | 62.97 | 82.17 | 80.49 | 87.44 | 49.82 | 84.75 | 53.66 | 87.6 | 65.56 | 84.23 |
| | Stable Diffusion 3 | 60.9 | 84.46 | 79.22 | 86.71 | 48.32 | 84.29 | 51.91 | 87.52 | 61.64 | 82.32 |
| | Stable Diffusion 3.5 Large | 60.16 | 83.91 | 80.53 | 89.22 | 48.93 | 85.33 | 51.53 | 88.69 | 63.21 | 83.79 |
| | Stable Diffusion 3.5 Large Turbo | 57.27 | 82.2 | 79.4 | 87.51 | 47.16 | 83.62 | 50.07 | 87.06 | 61.72 | 83.09 |
| | Flux.1 [dev] | 55.63 | 80.17 | 72.7 | 81.53 | 45.85 | 82.82 | 46.73 | 81.39 | 51.63 | 76.62 |
| | DALL-E Mini | 50.86 | 76.96 | 73.55 | 80.1 | 41.52 | 78.48 | 44.07 | 77.11 | 54.11 | 72.64 |
| | DALL-E 2 | 52.07 | 81.19 | 79.19 | 86.03 | 42.54 | 83.05 | 43.11 | 81.66 | 61.65 | 81.27 |
| | DALL-E 3 | 67.09 | 82.8 | 85.68 | 88.86 | 50.43 | 86.87 | 58.8 | 86.34 | 64.3 | 86.38 |
| | DALL-E 3 w/ Rewrite | 63.74 | 81.83 | 84.24 | 90.08 | 61.41 | 83.96 | 68.7 | 89.28 | 74.77 | 85.69 |
| | gpt-image-1 | 65.47 | 88.62 | 88.39 | 93.24 | 65.31 | 88.37 | 67.77 | 92.22 | 77.67 | 88.25 |
| Concise Prompts — T2V Models | Cosmos-1 | 59.61 | 76.57 | 68.87 | 76.26 | 53.27 | 72.08 | 56.84 | 78.34 | 54.04 | 65.18 |
| | Open-Sora | 65.46 | 84.56 | 75.56 | 83.21 | 48.49 | 85.21 | 59.79 | 87.59 | 59.19 | 80.56 |
| | VideoCrafter-2 | 61.3 | 82.13 | 76.19 | 84.12 | 42.9 | 86.43 | 53.3 | 88.76 | 61.73 | 83.51 |
| | CogVideoX | 36.72 | 59.54 | 42.55 | 55.8 | 27.71 | 61.82 | 28.76 | 63.23 | 25.98 | 44 |
| | Wan 2.1 | 29.57 | 62.49 | 47.02 | 68.41 | 30.37 | 54.07 | 30.68 | 65.81 | 30.23 | 67.89 |
| Detailed Prompts — T2I Models | Stable Diffusion 1.4 | 70.07 | 79.31 | 74.19 | 77.58 | 65.24 | 78.94 | 56.99 | 80.53 | 63.87 | 76.98 |
| | Stable Diffusion 1.5 | 71.03 | 79.97 | 73.5 | 77.69 | 65.21 | 78.89 | 56.84 | 79.72 | 63.02 | 77.06 |
| | Stable Diffusion XL | 72.82 | 84.76 | 80.84 | 85.6 | 61.1 | 85.85 | 61.1 | 87.44 | 70.93 | 83.55 |
| | Stable Diffusion 2.1 | 69.41 | 81.72 | 77.51 | 82.03 | 63.64 | 82.68 | 59.39 | 84.07 | 64.71 | 79.95 |
| | Stable Diffusion 3 | 74.27 | 87.11 | 82.58 | 88.48 | 66.21 | 86.95 | 62.59 | 88.08 | 67.32 | 83.13 |
| | Stable Diffusion 3.5 Large | 73.21 | 86.84 | 83.24 | 89.5 | 67.05 | 87.6 | 60.55 | 88.82 | 67.65 | 84.27 |
| | Stable Diffusion 3.5 Large Turbo | 73.24 | 86.23 | 81.07 | 88.24 | 64.83 | 86.37 | 58.46 | 87.81 | 65.05 | 82.98 |
| | Flux.1 [dev] | 72.86 | 85.56 | 77.43 | 85.19 | 61.47 | 82.72 | 58.52 | 85.31 | 59.56 | 79.66 |
| | DALL-E Mini | 53.69 | 74.12 | 69.5 | 73.38 | 52.39 | 72.11 | 50.47 | 73.65 | 58.22 | 68.92 |
| | DALL-E 2 | 64.72 | 79.34 | 80.2 | 85.79 | 62.33 | 83.66 | 55.51 | 82.6 | 66.07 | 80.34 |
| | DALL-E 3 | 77.75 | 85.3 | 83.82 | 87.99 | 68.16 | 86.26 | 71.19 | 87.79 | 73.51 | 84.35 |
| | DALL-E 3 w/ Rewrite | 76.73 | 87.12 | 85.56 | 90.33 | 76.36 | 85.43 | 75.63 | 91.22 | 78.8 | 86.54 |
| | gpt-image-1 | 78.26 | 90.7 | 86.88 | 90.94 | 79.47 | 88.85 | 78.04 | 92.86 | 79.39 | 88.38 |
| Detailed Prompts — T2V Models | Cosmos-1 | 64.61 | 72.64 | 67.1 | 73.94 | 57.62 | 67.03 | 56.83 | 73 | 50.39 | 58.99 |
| | Open-Sora | 74.81 | 86.48 | 76.69 | 80.84 | 67.65 | 83.93 | 69.59 | 86.77 | 71.15 | 81.49 |
| | VideoCrafter-2 | 70.88 | 85.37 | 79.53 | 83 | 66.14 | 87.21 | 62.58 | 86.47 | 68.14 | 83.72 |
| | CogVideoX | 39.83 | 50.63 | 46.4 | 54.35 | 38.89 | 57.82 | 35.8 | 62.68 | 25.51 | 40.11 |
| | Wan 2.1 | 55.79 | 79.96 | 62.14 | 73.08 | 42.39 | 73.82 | 48.86 | 76.6 | 48.73 | 75.8 |

**Applying the Mitigation Strategy to Text-to-Video Models**  While our proposed mitigation strategy is designed to be broadly compatible with most multimodal models, it would be interesting to apply our method to text-to-video generative models. Our experiments were limited to text-to-image models due to resource constraints. Therefore, we encourage future researchers with the necessary

Table 11: **Complete DialectGen Benchmark Performance Breakdown** by dialect for all text-to-image and text-to-video generative models. All performance scores are measured using CLIP-Score (Hessel et al., 2021), higher score is better. Results complements Table 2 in the main paper.

| | Model | Performance by Dialect (CLIPScore) ↑ | | | | | | | | | |
| | | AAE | | BrE | | ChE | | InE | | SgE | |
| | | Dialect | SAE | Dialect | SAE | Dialect | SAE | Dialect | SAE | Dialect | SAE |
| **Concise Prompts** — T2I Models | Stable Diffusion 1.4 | 25.46 | 27.83 | 28.65 | 29.79 | 24.68 | 28.34 | 24.34 | 29.38 | 25.97 | 28.64 |
| | Stable Diffusion 1.5 | 25.79 | 27.95 | 28.66 | 29.91 | 24.7 | 28.32 | 24.38 | 29.44 | 25.86 | 28.65 |
| | Stable Diffusion 2.1 | 25.72 | 28.74 | 29.67 | 30.88 | 24.7 | 29.44 | 25.31 | 30.69 | 26.46 | 29.54 |
| | Stable Diffusion XL | 25.81 | 28.69 | 29.97 | 31.21 | 25.37 | 29.57 | 25.85 | 31.15 | 27.45 | 30.23 |
| | Stable Diffusion 3 | 25.45 | 28.42 | 29.89 | 30.97 | 25.01 | 28.74 | 25.02 | 30.31 | 26.8 | 29.67 |
| | Stable Diffusion 3.5 Large | 25.62 | 28.78 | 30.25 | 31.5 | 25.22 | 29.42 | 25.67 | 31.14 | 27.18 | 30.22 |
| | Stable Diffusion 3.5 Large Turbo | 25.1 | 28.4 | 29.9 | 31.16 | 24.95 | 28.9 | 25.3 | 30.78 | 26.96 | 29.82 |
| | Flux.1 [dev] | 24.74 | 27.54 | 28.52 | 29.88 | 24.21 | 27.97 | 24.78 | 29.48 | 25.4 | 28.31 |
| | DALL-E Mini | 24.77 | 28.15 | 29.48 | 30.65 | 23.57 | 27.81 | 24.4 | 29.56 | 25.74 | 28.6 |
| | DALL-E 2 | 24.57 | 27.4 | 29.86 | 30.56 | 23.98 | 27.3 | 24.1 | 29.3 | 26.44 | 28.53 |
| | DALL-E 3 | 25.19 | 27.51 | 28.95 | 29.75 | 24.57 | 28.11 | 25.71 | 29.66 | 26.3 | 29.08 |
| | DALL-E 3 w/ Rewrite | 24.92 | 26.91 | 29.41 | 30.11 | 25.15 | 27.57 | 26.87 | 29.93 | 27.12 | 28.47 |
| | gpt-image-1 | 25.96 | 28.33 | 30.94 | 31.62 | 26.51 | 29.48 | 27.51 | 31.21 | 28.57 | 30.33 |
| T2V Models | Cosmos-1 | 23.49 | 25.42 | 26.17 | 27.16 | 22.89 | 24.62 | 24.18 | 27.04 | 21.89 | 22.91 |
| | Open-Sora | 25.02 | 27.3 | 28.63 | 29.73 | 24.34 | 27.09 | 25.35 | 29.36 | 25.55 | 28.01 |
| | VideoCrafter-2 | 25.88 | 28.83 | 29.41 | 30.69 | 25.04 | 29.04 | 25.88 | 30.56 | 27 | 29.69 |
| | CogVideoX | 22.62 | 25.71 | 24.14 | 25.84 | 22.03 | 24.61 | 22.95 | 27.4 | 19.99 | 22.18 |
| | Wan 2.1 | 22.37 | 25.49 | 25.45 | 28.27 | 22.14 | 24.55 | 22.55 | 27.57 | 22.75 | 26.85 |
| **Detailed Prompts** — T2I Models | Stable Diffusion 1.4 | 27.98 | 28.84 | 29.59 | 30.23 | 28.61 | 30.1 | 26.79 | 29.83 | 28.12 | 29.78 |
| | Stable Diffusion 1.5 | 28.08 | 28.99 | 29.54 | 30.29 | 28.52 | 30.18 | 26.87 | 29.94 | 27.98 | 29.82 |
| | Stable Diffusion 2.1 | 28.57 | 29.9 | 30.83 | 31.69 | 29.68 | 31.51 | 28.24 | 31.47 | 29.54 | 31.32 |
| | Stable Diffusion XL | 28.46 | 29.68 | 30.49 | 31.33 | 29.12 | 31.14 | 27.95 | 30.96 | 28.65 | 30.52 |
| | Stable Diffusion 3 | 28.69 | 29.82 | 30.76 | 31.59 | 29.13 | 31.15 | 27.82 | 31 | 29.13 | 31.04 |
| | Stable Diffusion 3.5 Large | 28.86 | 30.01 | 31.02 | 31.84 | 29.48 | 31.64 | 28.04 | 31.61 | 29.29 | 31.19 |
| | Stable Diffusion 3.5 Large Turbo | 28.61 | 29.58 | 30.67 | 31.6 | 29.09 | 31.23 | 27.8 | 31.18 | 28.76 | 30.78 |
| | Flux.1 [dev] | 27.97 | 28.69 | 29.54 | 30.37 | 28.17 | 30.17 | 27.15 | 30.01 | 27.72 | 29.46 |
| | DALL-E Mini | 27.26 | 29.18 | 29.84 | 30.56 | 27.75 | 30.23 | 26.71 | 29.93 | 27.42 | 29.59 |
| | DALL-E 2 | 27.66 | 29.02 | 30.5 | 31.3 | 28.57 | 30.54 | 26.48 | 30.17 | 28.69 | 29.88 |
| | DALL-E 3 | 27.79 | 28.3 | 29.55 | 30.21 | 28.52 | 30.04 | 27.48 | 29.83 | 28.67 | 30.03 |
| | DALL-E 3 w/ Rewrite | 27.71 | 28.23 | 29.75 | 30.46 | 28.88 | 29.85 | 28.57 | 29.98 | 28.61 | 29.42 |
| | gpt-image-1 | 28.65 | 29.45 | 31.2 | 31.6 | 29.98 | 31.29 | 29.41 | 30.81 | 30.18 | 31.27 |
| T2V Models | Cosmos-1 | 23.07 | 23.79 | 25.98 | 26.67 | 24.19 | 24.94 | 23.35 | 25.29 | 19.99 | 21.09 |
| | Open-Sora | 27.4 | 28.36 | 29.5 | 29.93 | 28.07 | 29.46 | 27.64 | 30.04 | 27.64 | 29.19 |
| | VideoCrafter-2 | 28.4 | 29.76 | 30.24 | 30.98 | 28.95 | 31 | 27.83 | 30.88 | 28.74 | 30.61 |
| | CogVideoX | 21.42 | 22.55 | 24.38 | 25.74 | 22.89 | 24.6 | 22.37 | 25.51 | 17.67 | 19.82 |
| | Wan 2.1 | 25.85 | 27.89 | 27.96 | 29.2 | 26.05 | 28.83 | 25.25 | 28.92 | 25.45 | 27.98 |

computing resources to experiment in this domain, as it would serve as a strong test of our method's generalizability.

