# OpenReview forum: "DialectGen: Benchmarking and Improving Dialect Robustness in Multimodal Generation"
_ICLR.cc/2026/Conference — Submitted to ICLR 2026_

### Official Review · Reviewer_UK5K · 2025-10-30

**Soundness:** 2
**Presentation:** 3
**Contribution:** 2
**Rating:** 4
**Confidence:** 5

**Summary:**

This paper introduces DialectGen, a benchmark evaluating dialect robustness in text-to-image and text-to-video generation. Moreover,  it proposes a method including three new loss functions to prevent significant dialect performance drops.

**Strengths:**

1. This paper is well-written and easy to follow
2. There is sufficient work related to benchmark construction and algorithm design.

**Weaknesses:**

The biggest issue lies in the contribution/value of the paper. There are many unresolved challenges in text-to-image and text-to-video models, such as visual aesthetics and the ability to adhere to complex prompts. However, prompt alignment primarily reflects whether the generative model can accurately simulate the objects and scenes described in the prompt, it measures generative capability rather than understanding (as many current models already use LLMs as text encoders to pursue stronger text understanding). For dialect prompts, the greater difficulty lies in feature extraction by the text encoder due to the lack of high-quality dialect training data. Therefore, I believe the dialect evaluation topic is more meaningful for assessing the understanding ability of the text encoder rather than the generative capability of the model. Assuming the availability of high-quality dialect data that enables the text encoder to effectively learn dialect features, I think simple dialect mismatch issues would no longer exist in generative models.

Thus, I consider the contribution of the paper to be limited, as it does not address the critical challenges of generative models. Instead, the issue of dialect understanding and adherence should be more appropriately tackled within the realm of language models.

**Questions:**

Please see Weakness. I hope these issues can be resolved, and I will reconsider my grading.

---

> ### Author Response · Authors · 2025-11-29
> **Responses to Reviewer UK5K (1/2)**
>
> We sincerely thank reviewer UK5K for their feedback. We are glad that you found:
>
> * the **paper well-written and easy to follow**, and
> * that we have contributed **sufficient work on benchmark construction and algorithm design**.
>
> We address your main concerns about the contribution and value of our work below.
>
> ---
>
> **On the contributions and scope of our work**
>
> We appreciate your comments on the value and scope of our contribution, and would like to clarify what we see as our main contributions:
>
> 1. **Identifying and accurately quantifying dialectal performance disparities in multimodal generation.**
>    The core contribution of our work is to **empirically identify and rigorously quantify** dialectal performance disparities in text-to-image and text-to-video generation. To our knowledge, no prior work has:
>    * systematically studied dialect robustness in multimodal generative models, or
>    * measured how a **single lexical dialect feature** can induce **32–48% performance drops** across many T2I/T2V systems.
>    While many challenges exist for generative models (aesthetics, long-horizon compositionality, etc.), dialect robustness is a **distinct, socially important failure mode** that has so far been unmeasured in multimodal generation.
>
> 2. **Providing DialectGen as a benchmark and resource for future studies.**
>    DialectGen is not only a dataset used in our experiments, but a **reusable benchmark and construction pipeline** for future models and methods. It provides:
>    * dialect-speaker–validated SAE–dialect prompt pairs with strict semantic equivalence, and
>    * a clear methodology for lexeme selection, prompt scaffolding, and human validation.
>    This benchmark makes it possible to **consistently evaluate** dialect robustness in multimodal generators going forward, much as prior work (e.g., VALUE / Multi-VALUE) did for text-only dialect tasks.
>
> Our goal is **not** to solve all “critical challenges” in T2I/T2V modeling, but to add a **new, rigorously defined axis of evaluation and mitigation** that has been previously missing.
>
> ---
>
> **Why dialect robustness is a multimodal problem, not only a text-encoder problem**
>
> We agree with the reviewer’s suggestion that dialect understanding failures can be primarily attributed to text encoder misalignment. However, we respectfully disagree with the implication that this makes dialect robustness *only* a language-model problem that should be studied separately from multimodal generative models:
>
> 1. **Harms manifest in the visual output used by real users.**
>    The **practical impact** of dialectal performance disparities in multimodal generation is that dialect-speaking users receive systematically worse or nonsensical images/videos. As discussed in the introduction:
>
>    > “Linguists have defined over 160 dialects within the English language, with three out of four English speakers having a dialect background other than Standard American or British English … Prior works have shown significant allocational harms toward dialect speakers caused by such dialect performance discrepancies in machine learning applications … making the observation of similar performance trends in multimodal generative models an alarming sign.” (L48–55)
>
>    Today’s multimodal generative models are already **widely deployed and used directly by dialect speakers**. Evaluating only the text encoder in isolation would miss how misinterpretations affect the **actual images and videos** people consume.
>
> 2. **End-to-end behavior with dialect inputs is what matters for fairness.**
>    Our evaluation metrics measure whether a user who writes a dialect prompt receives an image/video that is **semantically equivalent** to what they would get with the SAE version of the same request. This is precisely the behavior that matters for **allocational and representational fairness** in current deployed multimodal generative models.
>
> 3. **Better encoders alone have not solved the problem in practice.**
>    We agree in theory that, with abundant high-quality dialect data, text encoders *should* learn better dialect features. However, empirically:
>    * Current pre-training pipelines **filter out** many lower-resource dialects from web-scale data, “excluding data involving lower-resource English dialects other than Standard American and British English … reducing the effectiveness of pretrained models on inputs from other dialects” (L48–55).
>    * Several of the models we evaluate already employ **strong LLM-based or large CLIP-style encoders**, yet still exhibit **32–48% performance drops** on dialect prompts.
>    This indicates that increased encoder capacity and current data alone have **not** eliminated dialect mismatch in deployed generative systems.
>
> For these reasons, we believe it is both appropriate and necessary to study dialect robustness **in the multimodal generation setting itself**, while still addressing the encoder as the locus of the mitigation.

---

> ### Author Response · Authors · 2025-11-29
> **Responses to Reviewer UK5K (2/2)**
>
> **On the assumption that “good dialect data will solve it” and that this should only be tackled in LMs**
>
> The review suggests that, given high-quality dialect data for the text encoder, “simple dialect mismatch issues would no longer exist in generative models,” and that dialect understanding should be addressed within language models instead of generative models.
>
> We respectfully see this as an **idealized assumption** that is unlikely to hold in practice, and potentially problematic if taken as a reason *not* to act on multimodal harms now:
>
> 1. **Data imbalance and filtering are structural, long-term issues.**
>    For many dialects (AAE, Singlish, etc.), large, high-quality, ethically sourced corpora comparable to mainstream SAE are difficult to obtain, and pre-training pipelines explicitly **filter out** such data. Even with ongoing efforts, we are unlikely to reach a state of “perfect dialect coverage” in the near future, and data biases will **continue to persist**.
>
> 2. **We cannot wait for perfect text-side solutions before mitigating multimodal harms.**
>    Multimodal generative systems are already being **deployed at scale** and used daily by dialect speakers around the world. Deferring multimodal fairness until the text side is “fully solved” would mean accepting years of **avoidable allocational harms** in image/video generation. By the same logic, many AI safety and fairness issues could be dismissed with “given good data all this can be solved,” but empirically that has not been the case.
>
> 3. **Our work explicitly tackles the encoder problem in the multimodal context.**
>    Rather than ignoring the text-encoder root cause, our mitigation method **directly operates on the encoder**:
>    * It improves dialect representations using a modest amount of curated dialect data.
>    * It preserves SAE and polysemy behavior via Polysemy Control and KL Regularization.
>    In this sense, our work bridges the LM and multimodal perspectives: we treat the encoder as the locus of learning, but we **evaluate and optimize it with respect to the actual multimodal behavior** that users experience.
>
> ---
>
> **Role of DialectGen as standardized multimodal evaluation**
>
> Finally, we would like to emphasize the role of DialectGen even under future improvements in encoder training and data:
>
> * A benchmark like DialectGen will remain important to:
>   * verify whether improvements are **uniform across dialects and modalities**, and
>   * detect regressions or unintended trade-offs (e.g., better dialect alignment but worse polysemy or compositionality).
> * DialectGen also provides a platform to study deeper questions, such as whether **multimodal training introduces additional failure modes** beyond what is seen in text-only models. Our current work does not claim to have resolved those questions, but it provides the **measurement infrastructure** needed to investigate them.
>
> We will revise the introduction and discussion to more clearly state that our primary contribution is to **identify and accurately quantify multimodal dialect disparities and to provide a benchmark that enables their systematic study and mitigation**, while our encoder-based method demonstrates that these disparities can be significantly reduced in currently deployed systems without retraining the entire generative model.

---

### Official Review · Reviewer_nKia · 2025-10-31

**Soundness:** 3
**Presentation:** 3
**Contribution:** 2
**Rating:** 4
**Confidence:** 3

**Summary:**

This paper constructs DialectGen, a large-scale benchmark for multimodal generative models’ dialect robustness, covering 6 English dialects (e.g., SAE, AAE, InE) with 4,200 validated SAE-dialect prompt pairs. Evaluating 17 T2I/T2V models (e.g., Stable Diffusion, DALL-E), the authors find a single dialect lexical feature causes 32.26%–48.17% performance degradation, with T2V models like Wan 2.1 suffering the worst drops. Existing methods (e.g., UNet fine-tuning) only improve dialect performance by <7% and harm SAE results, so the authors propose an encoder-based strategy with three losses. This method raises 5 dialects’ performance to match SAE (+34.4% on average) with near-zero SAE loss (<1%) on Stable Diffusion 1.5/XL.

**Strengths:**

1. DialectGen targets lexical variations (shown to drive >25% performance drops, vs. <2% for grammatical variations) and is built via authoritative regional dictionaries, filtering of derogatory/culture-unique terms, and validation by dialect speakers (35.9% of prompts rejected), ensuring high reliability.

2. The authors assess 17 models across two prompt settings and three metrics (VQAScore, CLIPScore, human evaluation).

3. The encoder-based strategy avoids the SAE performance loss of baselines. Ablation studies confirm each loss component’s value.

**Weaknesses:**

1. DialectGen includes only 6 English dialects, omitting low-resource varieties (e.g., Caribbean English, Australian English) that are more vulnerable to allocational harms, limiting the benchmark’s global generalizability.

2. While the paper notes Prompt Revision (e.g., GPT4.1 Prompt Translate) only improves dialect performance by up to 6.1%, it lacks detailed analysis of why this method fails—especially given that standalone LLM translation (e.g., GPT-4o translating dialect words to SAE) may appear effective in isolated tests. No failure cases or contextual constraints (e.g., concise prompts) are discussed.

3. The proposed encoder-based method is only tested on Stable Diffusion 1.5 and SDXL (T2I models). T2V models (which suffer the worst performance drops) and proprietary/non-diffusion architectures (e.g., DALL-E 3, Open-Sora) are unaddressed, limiting the method’s real-world applicability. In contrast, Prompt Revision exhibits greater flexibility: it relies solely on general-purpose LLMs (e.g., GPT4.1, LLaMA 3) to rewrite or translate input prompts. This makes it easily applicable to proprietary models like DALL-E 3, as it does not depend on model developers providing fine-tuning interfaces or exposing internal architectural details.

**Questions:**

1. You note Prompt Revision methods (e.g., GPT4.1 Prompt Translate) only improve dialect performance by up to 6.1%, yet standalone tests (e.g., using GPT-4o with “Translate this sentence into standard English”) can accurately replace dialect words with SAE equivalents. Could you explain why Prompt Revision underperforms in your experiments? Please provide specific failure cases (e.g., prompts where revision failed to replace dialect lexemes, or revised prompts still led to poor model generation) and analyze contextual factors.

2. GPT4o was used to generate DialectGen’s prompts, which may introduce SAE-centric biases (e.g., overrepresenting Western contexts). Did you validate that these prompts reflect natural dialect usage (e.g., comparing to real-world dialect text from social media or regional corpora) or confirm with dialect speakers that prompts align with how they would naturally phrase the same scenes (e.g., for “carnal” in ChE)?

3. Your KL Regularization uses MSCOCO, an SAE-centric image-caption dataset. Could this reinforce biases in dialect generations—for example, making “ang pow” (SgE) look more like Western “red packets” than culturally accurate Singaporean red envelopes? Have you tested region-specific datasets (e.g., Singaporean food/image corpora for SgE) for KL Regularization, and if so, how did performance and cultural alignment change?

---

> ### Author Response · Authors · 2025-11-30
> **Responses to Reviewer nKia (1/3)**
>
> We sincerely thank the reviewer for the thoughtful and careful assessment of our work. We are glad that you highlighted several important strengths of the paper, including:
>
> - the **high reliability** of DialectGen’s construction, noting its use of authoritative regional dictionaries, filtering of derogatory or culture-unique terms, and **rigorous dialect-speaker validation**, where *35.9% of prompts were rejected* to ensure correctness.
> - the **comprehensive evaluation** of *17 T2I/T2V models* across two prompt settings and three metrics (*VQAScore, CLIPScore, human evaluation*).
> - the **effectiveness of our encoder-based mitigation strategy**, with ablations confirming the value of each loss component and its ability to improve dialect robustness while maintaining **near-zero SAE performance loss**.
>
> We address the reviewer’s concerns and questions below.
>
> ---
> **Dialect coverage and benchmark generalizability**
>
> We fully agree with the reviewer that expanding to more English varieties (e.g., Caribbean English, Australian English) would further increase the benchmark’s global breadth. In this first iteration, however, we deliberately prioritize **depth and reliability over breadth**, consistent with prior work in cross-dialectal evaluation ([1], [2]):
>
> 1. **Prioritizing high data quality and fair annotation practices**
> DialectGen emphasizes high-quality, culturally grounded data rather than maximizing dialect count.
> * We curate **4,200+ dialect prompts** across six English dialects using authoritative regional dictionaries.
> * Each SAE–dialect pair is validated by **at least three dialect-speaker annotators** for fluency and exact semantic equivalence, resulting in a substantial rejection rate.
> * Annotators are **fairly compensated**, and given our limited annotation budget, this level of validation makes each data point expensive but significantly improves reliability.
>
> 2. **Designed from the start for extensibility and reproducibility**
> As the first work to study dialect robustness in multimodal generation, our goal is to provide a **reliable, extensible foundation** rather than exhaustive dialect coverage.
> * We explicitly document every step of the construction pipeline—lexeme selection, prompt scaffolding, filtering, and multi-annotator validation—so researchers can straightforwardly add new dialects or languages.
> * This mirrors the strategy of VALUE and Multi-VALUE ([1], [2]), which first validated their methodology on a small set of dialects before expanding to broader linguistic coverage in follow-up works.
>
> 3. **Coverage of diverse linguistic and sociocultural settings**
> Even with six dialects, DialectGen spans a wide and meaningful range of English varieties—AAE, ChE, InE, SgE, BrE, SAE—covering:
> * multiple continents and sociolinguistic histories,
> * both high-resource and lower-resource dialects, and
> * varieties known to experience allocational harms.
> This diversity is sufficient to show the **universality of dialect-induced failures** and to motivate broader expansions in future work.
>
> [1] VALUE: Understanding Dialect Disparity in NLU (ACL 2022)
> [2] Multi-VALUE: A Framework for Cross-Dialectal English NLP (ACL 2023)

---

> ### Author Response · Authors · 2025-11-30
> **Responses to Reviewer nKia (2/3)**
>
> **Why Prompt Revision underperforms despite strong standalone LLM translation**
>
> We appreciate this question and agree that LLMs like GPT-4o can often translate individual dialect sentences into SAE. However, in **end-to-end multimodal generation**, Prompt Revision faces structural and empirical limitations that explain its modest gains (≤ 6.1%):
>
> 1. **Prompt Revision must be applied blindly in deployment**
>    Since real systems cannot detect which dialect or language a prompt belongs to, rewriting must be applied **uniformly to all prompts**, including SAE prompts.
>    This introduces a **compounding error source**, unnecessary semantic drift, and added compute overhead. Even correct SAE translations may still fall victim to SAE-centric visual priors in generative models.
>
> 2. **Translation errors from automatic “correction”**
>
> | Category | Dialect Prompt | Intended SAE (✅) | LLM Translate (❌) |
> |---------|----------------|-------------------|---------------------|
> | **Mis-correction** | `a photo of the matha` | `a photo of the monastery` | `a photo of the math` |
>
> The LLM mistakenly treats *matha* as a typo, producing an incorrect rewrite.
>
> 3. **Polishing tone but failing to translate the key lexeme**
>
> | Category | Dialect Prompt | Intended SAE (✅) | LLM Translate (❌) |
> |---------|----------------|-------------------|---------------------|
> | **Tone polishing without lexical translation** | `a woman getting help from the dai` | `a woman receiving help from the midwife` | `a woman receiving help from the dai` |
>
> The LLL refines phrasing but leaves the dialect lexeme untouched, preserving the original error.
>
> 4. **Polysemy and context-selection errors**
>
> | Category | Dialect Prompt | Intended SAE (✅) | LLM Translate (❌) |
> |---------|----------------|-------------------|---------------------|
> | **Polysemy (Singlish “blur”)** | `a guy feeling blur at the front of the class` | `a guy feeling confused at the front of the class` | `a blurry guy at the front of the class` |
>
> LLM incorrectly selects the **visual** sense (“blurry”), not the dialect sense (“confused”), causing systematically wrong generations.
>
> 5. **Concise prompts are brittle to paraphrasing (including DALLE-3-style rewriting)**
>    Even small changes in ≤ 6-word prompts significantly alter what models generate.
>
> **(a) Over-augmentation)**
>
> | Dialect Prompt | Intended SAE (✅) | LLM Rewrite (❌) |
> |----------------|-------------------|------------------|
> | `two kids opening ang pow` | `two kids opening red packets` | `two happy children opening traditional red envelopes for Chinese New Year` |
>
> Added festival context shifts the model toward lion dance, fireworks, and crowd scenes.
>
> **(b) Semantic drift**
>
> | Dialect Prompt | Intended SAE (✅) | LLM Rewrite (❌) |
> |----------------|-------------------|------------------|
> | `a boy eating pani puri` | `a boy eating pani puri` | `a boy enjoying spicy Indian street food pani puri` |
>
> Adjectives like “spicy” and “street food” lead to unintended stalls or busy outdoor scenes.
>
> 6. **Systematic evaluation reveals cumulative degradation**
> Across thousands of prompts, issues such as drift, mis-correction, polysemy errors, and untranslated lexemes accumulate, resulting in only modest overall improvement.
> Our encoder-based method avoids these compounding issues by aligning dialect semantics **directly in embedding space**, while preserving SAE behavior.
>
> We will include representative failure cases and concise qualitative analysis in the revised manuscript.

---

> ### Author Response · Authors · 2025-11-30
> **Responses to Reviewer nKia (3/3)**
>
> **On the real-world applicability of our mitigation strategy, and compared with prompt rewriting**
>
> 1. **Applicability to text-to-video models and why T2V experiments were not included**
>
> We fully agree with the reviewer that extending our mitigation to text-to-video (T2V) generators is an important next step. Our mitigation strategy was **explicitly designed to generalize** to any multimodal generative system that contains a **text encoder producing embeddings**, which is a universal component across modern T2I and T2V models.
>
> To support this, we apply our method to **two open-source T2I models with substantially different encoder architectures**:
> * **Stable Diffusion 1.5**, which uses a *single* text encoder, and
> * **Stable Diffusion XL**, which uses a *dual* base–refiner encoder stack.
>
> These two architectures span much of the diversity seen in modern T2I/T2V systems, and the strong gains on both confirm that our method is **architecture-agnostic** and directly applicable to any model that exposes its encoder.
>
> The reason we did **not** include full T2V mitigation experiments is purely **computational**. Running our SD1.5 experimental protocol on a relatively lightweight open-source T2V model such as VideoCrafter—same number of prompts, seeds, training steps, and evaluation frames—would require approximately **85,000 A6000 GPU hours**, due to the need to generate and evaluate multiple frames per sample. This level of compute far exceeds what an academic research lab can access.
>
> Given this constraint, our goal in this work was to:
>
> * demonstrate using DialectGen that **T2V models suffer the most severe dialect drops**,
> * and show that our encoder-based method succeeds across **two architecturally distinct T2I pipelines**, making it immediately applicable *in principle* to T2V models that reuse their text encoder (which all current T2V pipelines do).
>
> We will clarify this design choice and compute limitation in the revised manuscript.
>
> ---
>
> 2. **Prompt rewriting is not inherently “more flexible” for proprietary models**
>
> We appreciate the reviewer for raising this comparison. To clarify:
> It is **not** that our method is limited in applicability—it is the **access level** provided by proprietary model vendors that limits where the method can be externally applied.
>
> Our mitigation requires only one capability:
> **access to the model’s text encoder for alignment training.**
> This module exists in all modern multimodal generative systems, including diffusion and autoregressive architectures.
>
> If proprietary providers chose to update or expose their text encoders (as is done routinely for safety alignment, RLHF, and embedding updates), they could apply our **one time mitigation** internally and offer users:
>
> * **dialect-robust generation at no added inference cost**,
> * **no LLM rewriting latency**,
> * **no memory or throughput penalties**, and
> * **no compounding paraphrasing errors**.
>
> By contrast, Prompt Revision methods:
>
> * must be run **at inference time for every prompt**,
> * must be applied **blindly** to both SAE and dialect prompts (since dialect cannot be reliably detected),
> * introduce direct **semantic drift**, **mis-correction**, and **sense-selection errors**,
> * and impose **non-trivial compute overhead** on end users or API providers.
>
> Thus, if providers implemented our encoder-level adjustments internally (a modest training task), they could deliver **dialectally robust endpoints** that outperform prompt rewriting while being **cheaper, faster, and more stable** at inference time.

---

### Official Review · Reviewer_cLvx · 2025-10-31

**Soundness:** 3
**Presentation:** 3
**Contribution:** 2
**Rating:** 6
**Confidence:** 3

**Summary:**

This paper tackles a critical issue: generative models fail when users input dialectal English (like Singlish or AAE). The authors introduce "DialectGen," a high-quality benchmark built with native speakers, to prove this performance drop (up to 48%). They also propose a smart encoder-tuning method that teaches models new dialect words (e.g., "whip" = "car") without making them forget the original meaning ("whip" = "lash") or hurting general performance.

**Strengths:**

Important Problem: This is a timely and crucial problem. As models go global, supporting all users, not just those speaking standard English, is essential.

Excellent Benchmark: The "DialectGen" dataset is a significant contribution. Using native speakers for validation is the right way to do this, and the paired (SAE vs. Dialect) design is perfect for isolating the problem.

Well-Designed Method: The mitigation strategy is clever. The three-part loss function (Dialect Learning, Polysemy Control, KL Reg) is an elegant way to add new knowledge while protecting existing abilities. The results (fixing dialect performance with <1% hit to SAE) are excellent.

**Weaknesses:**

Video Models Untested: The fix was only applied to image models (SD 1.5, SDXL). The paper shows video models suffer even more from this problem, so it's a clear gap not to test the solution on them, even if it was just due to compute constraints.

Compositionality: The benchmark tests single-word swaps. It's unclear how the models would handle prompts with multiple dialect words or even mixed dialects.

**Questions:**

See the weakness. I found this task quite interesting.

---

> ### Author Response · Authors · 2025-11-27
> **Responses to Reviewer cLvx**
>
> We sincerely thank reviewer **cLvx** for the thoughtful and encouraging feedback. We are glad that you found:
>
> * our work addresses an **important and timely problem** as models go global,
> * the **DialectGen benchmark a significant contribution**, with native-speaker validation and paired SAE–dialect design, and
> * our **mitigation strategy well-designed**, with the three-part loss (Dialect Learning, Polysemy Control, KL Reg) providing an elegant way to add dialect knowledge while preserving existing abilities.
>
> We address your concerns about video models and compositionality below.
>
> ---
>
> **Mitigation experiments on text-to-video models**
>
> We fully agree with the reviewer that extending our mitigation to text-to-video models is an important next step. In fact, our mitigation strategy was intentionally designed with this **generalization in mind**, assuming only a **text encoder that produces embeddings** (a common module across all current text-to-image and text-to-video generative models). We further experiment with two open-source models with vastly different architecture to show the generalizability of our method. In the paper, we apply the method to **two open-source T2I models with markedly different architectures** (SD1.5 with a single text encoder and SDXL with dual text encoders) precisely to demonstrate this architectural generality.
>
> While it would indeed be ideal to apply the same mitigation to T2V models, the **computational cost** is beyond what we can realistically access as an academic lab. Based on our estimate, replicating our SD1.5 experiments in Table 3 on a relatively small text to video generative model like VideoCrafter (same number of epochs, prompts, and evaluation protocol) would require on the order of **85,000 A6000 GPU hours**, due to the need to generate and score multiple frames per prompt.
>
> Given this constraint, we chose to:
>
> * use DialectGen to **show the problem exists and is severe** for both T2I and T2V models, and
> * demonstrate that our **encoder-based mitigation works across substantially different T2I architectures**, making it directly applicable in principle to T2V models that share the same text encoder module.
>
> We will clarify this design and the compute limitation more explicitly in the revision.
>
> ---
>
> **Compositional effects of multiple dialect words**
>
> We are glad that the reviewer notices this point from our limitations section. We would like to take the opportunity to share some additional insights and a potential future direction:
>
> We fully agree with the reviewer that having multiple dialect words in a sentence could be an even more challenging scenario for multimodal generative models. Given that the current SOTA models already exhibit large performance drops when a single dialect word is used, and the challenges associated with constructing such a high-quality dataset, we believe it would be an appropriate topic for future work.
>
> Here we provide a small quantitative experiment on 100 newly annotated SAE-dialect prompt pairs from the Indian English setting. Each prompt contains 2-3 SAE words with dialect word equivalents. In the **Single Dialect Word** setting, we replace one of them with its dialect word equivalent, and in the **Multi Dialect Word** setting we replace 2-3 words. Finally, we evaluate each model’s generation quality with the VQAScore metric (similar to Table 5 in our paper):
>
> | Model                      | SAE Prompt Performance | Single Dialect Word Prompt Performance | Multi Dialect Word Prompt Performance |
> |----------------------------|------------------------|----------------------------------------|----------------------------------------|
> | DALL·E Mini                | 72.39                  | 48.65 (-32.79%)                        | 32.47 (-55.15%)                        |
> | FLUX.1 dev                 | 83.04                  | 54.81 (-34.00%)                        | 38.90 (-46.26%)                        |
> | Stable Diffusion 3.5 Large | 88.73                  | 56.21 (-36.65%)                        | 37.62 (-57.60%)                        |
>
> Please note this is only an exploratory experiment and data quality is not guaranteed by rigorous dialect speaker validation.

---

### Official Review · Reviewer_F7tQ · 2025-11-01

**Soundness:** 2
**Presentation:** 2
**Contribution:** 2
**Rating:** 4
**Confidence:** 4

**Summary:**

The authors introduce DialectGen, a benchmark for evaluating dialect robustness in multimodal generation across six english dialects.

**Strengths:**

- the analysis and modeling on english dialect data is rarely explored
- the dataset is carefully designed with dialect speaker validation and semantic equivalence filtering
- over 17 models are evaluated, using both automatic metrics and human judgement

**Weaknesses:**

-  the proposed mitigation approach is a combination of standard alignment objectives, lacking deeper theoretical or architectural innovation
- only six dialects are included, which restricts generalizability and reduces potential community impact; it would be more convincing if extended to more dialects or other languages.
- the main advance lies in dataset construction and fine-tuning; while valuable empirically, the contribution is incremental and limited

**Questions:**

Please refer to the weakness section

---

> ### Author Response · Authors · 2025-11-27
> **Responses to Reviewer F7tQ (1/2)**
>
> We sincerely thank reviewer F7tQ for their feedback. We are glad that you found:
>
> * our **analysis and modeling on English dialect data as addressing an under-explored research problem**,
> * our dataset **carefully designed with dialect-speaker validation and semantic equivalence filtering**, and
> * our evaluation **covers over 17 models using both automatic metrics and human judgement**.
>
> We address your concerns about the mitigation approach, dialect coverage, and overall contribution below.
>
> ---
>
> **On the contributions and novelty of our work**
>
> We appreciate your comment on the overall contribution of our work, and would like to use this opportunity to clarify our main contributions and novelty:
>
> 1. The most important contribution of our work is to **identify and accurately quantify** dialectal performance disparities in text-to-image and text-to-video generation. To our knowledge, no prior work has:
>    * systematically studied dialect robustness in multimodal generative models, or
>    * quantified how a **single lexical dialect feature** can induce **32–48% performance drops** across a wide range of T2I/T2V models.
>    Dialect harms have been studied in NLU, but not in **multimodal generation**, where the harms are directly visible in images and videos. We see making this problem visible and measurable as the central novelty of our paper.
>
> 2. DialectGen includes a **reusable diagnostic benchmark and a robust dataset construction pipeline**, providing:
>    * a high-quality, dialect-speaker–validated resource for evaluating future models, and
>    * a **clear, reproducible pipeline** (dictionary-based lexeme selection, LLM scaffolding only as proposal, strict dialect-speaker filtering) that makes it easier to scale up to more dialects and languages.
>    This is analogous in spirit to works like VALUE [1] and Multi-VALUE [2] in the text-only setting: we define **what to measure** and **how to construct reliable data** for dialect disparity, here in the multimodal domain. We hope this will serve as a template that can be extended by us and others.
>
> 3. **A practical mitigation recipe with strong empirical gains.**
>    Our encoder-based method is intentionally simple yet effective. It:
>    * operates **only on the text encoder**, leaving the generator unchanged,
>    * is **data- and compute-efficient**, and
>    * yields substantial improvements on five dialects (up to +34.4%) with **< 1% SAE degradation** and preserved polysemy behavior.
>    This demonstrates that the disparities we expose are not only observable, but also **actionable with modest resources** once they are properly diagnosed.
>
> We will clarify this hierarchy of contributions in the introduction and conclusion so that the central novelty of **identifying and quantifying multimodal dialect disparities via DialectGen** is made explicit, with our mitigation method as a practical response.

---

> ### Author Response · Authors · 2025-11-27
> **Responses to Reviewer F7tQ (2/2)**
>
> **On the design of our mitigation approach and alignment objectives**
>
> We would like to clarify that our goal on the mitigation modeling side is to provide an **adequate, robust, and deployable solution** to the specific failure mode we uncover, rather than to introduce a new architecture or heavy theory. We view the simplicity and effectiveness of our approach as a strength for adaptation:
>
> * It is **encoder-only**, which is crucial when the diffusion/video backbone is proprietary or extremely expensive to retrain.
> * It is **lightweight and architecture-agnostic**, which makes real-world adoption more feasible.
>
> Within this simple framework, the losses are tailored specifically to multimodal dialect robustness:
>
> * **Dialect Learning** directly closes the lexical gap between dialect and SAE prompts in the encoder space.
> * **Polysemy Control** prevents regressions on polysemous words with both dialect and SAE meanings, an important issue for deployed systems.
> * **KL Regularization over similarity distributions** stabilizes the encoder on SAE captions so that existing generation quality is preserved while dialect robustness improves.
>
> We will clarify these design motivations and emphasize that the contribution here is a **purpose-built, simple alignment scheme** that works well in practice for the problem we expose.
>
> ---
>
> **On the number of dialects evaluated and the scope of DialectGen**
>
> We fully agree with the reviewer that extending to more dialects and other languages would further increase impact. In this first iteration, we are constrained by **annotation budget** and the high cost of obtaining *reliable* dialect data, so we deliberately prioritize **depth and quality over breadth**:
>
> * We curate over **4,200 unique dialect prompts** across six English dialects, sourced from authoritative regional dictionaries.
> * Each SAE–dialect pair is carefully checked by **at least three dialect-speaker annotators** for fluency and exact semantic equivalence, with a substantial rejection rate, and annotators are compensated at fair rates. This makes each data point expensive, but greatly improves reliability.
>
> At the same time, we design DialectGen and its construction pipeline to be **extensible**:
>
> * The methodology (lexeme selection, prompt scaffolding, multi-annotator validation) is clearly specified, making it straightforward for future work (by us or others) to add more English varieties or other languages.
> * Previously established research ([1], [2]) has also mirrored our strategy of first validating a successful data collection and benchmarking strategy on a smaller number of dialects before scaling up to many more in follow-up works.
> * The current six dialects (AAE, ChE, InE, SgE, BrE, SAE) already span multiple regions, sociohistorical backgrounds, and levels of resourcing and stigma, and we observe **consistent large performance drops** across them.
>
> We will make this extensible, “benchmark-first” positioning explicit, emphasizing that DialectGen provides a **high-quality foundation and a clear recipe** for scaling to more dialects as community resources and regional expertise become available.
>
> [1] VALUE: Understanding Dialect Disparity in NLU (ACL 2022)
> [2] Multi-VALUE: A Framework for Cross-Dialectal English NLP (ACL 2023)

---

### Author Response · Authors · 2025-12-03
**General Response and Summary of Reviews**

Dear AC and SAC,

We sincerely thank you for your service, especially in light of the recent developments. For your convenience, we summarized our paper's strengths as highlighted by reviewers:

* **Significance and Novelty**: DialectGen tackles the timely and crucial problem (`cLvx`) of dialectal failures in multimodal generation, a space that is rarely explored (`F7tQ`) yet essential to users around the globe (`cLvx`).

* **Benchmark Quality**: DialectGen is a high-quality (`cLvx`), carefully designed benchmark (`F7tQ`), built with native-speaker validation, semantic-equivalence filtering, and authoritative lexical resources, ensuring high reliability (`nKia`).

* **Benchmark Insights**: The benchmark’s paired SAE–dialect design provides clean causal identification of performance gaps, revealing substantial degradation from a single lexical feature (`cLvx`, `nKia`).

* **Extensive Evaluation**: The evaluation is extensive, covering 17 T2I/T2V models, multiple prompt settings, and both automatic and human metrics (`F7tQ`, `nKia`).

* **Well-designed Mitigation**: The proposed mitigation strategy is well-designed, featuring a clever three-part loss that enables dialect learning while preserving polysemy understanding and general SAE performance (`cLvx`).

* **Strong Mitigation Performance**: Experimental results demonstrate large improvements on dialect inputs with near-zero SAE regression, outperforming prior fine-tuning approaches while avoiding their degradation issues (`cLvx`, `nKia`).

* **Clear Presentation**: The paper is well-written, easy to follow, and presents sufficient work in both benchmark construction and algorithm design (`UK5K`).

We have also addressed each reviewer's initial reservations in detail below. We hope you find our answers satisfactory, as recent developments have prevented reviewers from responding or updating their scores.

Best regards,

Authors

---

### Meta-Review · Area_Chair_rkXF · 2026-01-25

**Summary:**

The submission addresses an underexplored problem of dialect robustness in multimodal generation. While DialectGen makes a valuable empirical contribution to identifying dialect disparities in multimodal generation, the authors’ responses to core reviewer concerns are not convincing enough. The unresolved issues—particularly around the contribution of the work and generalizability to T2V models—undermine the work’s rigor and real-world applicability. For these reasons, I recommend rejecting the submission.

**Reviewer Concerns:**

Addressed Concerns:
- The authors provided empirical justification for focusing on 6 English dialects and documented an extensible pipeline, responding to Reviewer F7tQ's and nKia's feedback on benchmark generalizability.
- The authors provided experimental results of the compositional effects of multiple dialect words, responding to Reviewer cLvx's concern.



Outstanding Concerns:
- The contribution and value of the work raised by Reviewer F7tQ and UK5K.
- The generalizability to T2V models raised by Reviewer cLvx and nKia.

**Reviewer Scores:**

- Reviewer F7tQ (Initial: 4): Maintains score due to unresolved concerns about the contribution of the work.
- Reviewer cLvx (Initial: 6): Lowers to 5. Computational constraints are reasonable, but the lack of T2V validation undermines real-world generalizability.
- Reviewer nKia (Initial: 4): Maintains score. The real-world applicability remains unresolved.
- Reviewer UK5K (Initial: 4): Maintains score. The concerns about the contribution and value of the work remain outstanding.

---

### Decision · Program_Chairs · 2026-01-26

Reject